# Light- and bias-induced structural variations in metal halide perovskites

Dohyung Kim[1], Jae Sung Yun[2], Pankaj Sharma[1], Da Seul Lee[2], Jincheol Kim[2], Arman M. Soufiani[2], Shujuan Huang[2], Martin A. Green[2], Anita W.Y. Ho-Baillie [2] & Jan Seidel[1]

Organic–inorganic metal halide perovskites have gained considerable attention for next-generation photovoltaic cells due to rapid improvement in power conversion efficiencies. However, fundamental understanding of underlying mechanisms related to light- and bias-induced effects at the nanoscale is still required. Here, structural variations of the perovskites induced by light and bias are systematically investigated using scanning probe microscopy techniques. We show that periodically striped ferroelastic domains, spacing between 40 to 350 nm, exist within grains and can be modulated significantly under illumination as well as by electric bias. Williamson-Hall analysis of X-ray diffraction results shows that strain disorder is induced by these applied external stimuli. We show evidence that the structural emergence of domains can provide transfer pathways for holes to a hole transport layer with positive bias. Our findings point to potential origins of $I$–$V$ hysteresis in halide perovskite solar cells.

[1] School of Materials Science and Engineering, University of New South Wales, Sydney, NSW 2052, Australia. [2] Australian Centre for Advanced Photovoltaics (ACAP), School of Photovoltaic and Renewable and Engineering, University of New South Wales, Sydney, NSW 2052, Australia. These authors contributed equally: Dohyung Kim, Jae Sung Yun. Correspondence and requests for materials should be addressed to A.W.Y.H.-B. (email: a.ho-baillie@unsw.edu.au) or to J.S. (email: jan.seidel@unsw.edu.au)

Organic–inorganic perovskite solar cells have undergone rapid improvements in energy conversion efficiencies increasing from 3.8% to 22.7% within just a few years[1–5]. Self-assembling organic–inorganic metal halide perovskite materials have been shown to have excellent photovoltaic properties such as large absorption coefficients, ease of band-gap tuning, high carrier mobilities, high carrier diffusion lengths, and direct band gaps[1,6–8]. However, devices fabricated using these materials still exhibit hysteresis in current and voltage (I–V) and transient characteristics which have been widely reported[9–16]. It is observed that the sweeping voltage direction and the rate affect the I–V characteristics greatly[11–13,17]. There have been numerous efforts to elucidate the underlying mechanisms, i.e., to clarify the origin of the hysteretic behaviour[11–13,17]. It has been experimentally and theoretically demonstrated that migration of ionic defects within the film, especially at the grain boundaries, is one of the most prominent causes for the observed transient hysteretic behaviour[18–21]. Ferroelectric polarization is also presumed to be a possible origin of the hysteresis, which can arise either from the orientation of the organic (dipolar) cations, or from deformation of the inorganic framework[22]. The dipole moment of A-site organic molecules can be rotated by an external electric field producing a balancing field that mitigates the effective field acting on the charge carriers. Although many theoretical studies using first principles calculations affirm the spontaneous polarization in halide perovskites, experimental results often face challenges in interpretation, as the material typically shows a large amount of leakage current and dynamic movement of ionic defects[16,23,24], which can lead to artefacts and inconclusive results.

Earlier investigation using piezoresponse force microscopy (PFM) measurements on MAPbI$_3$ perovskite showed switchable ferroelectric domains that are approximately equal in size to the grain[25]. Nevertheless, strong evidence related to domain and wall motion has not been provided, although hysteretic PFM loops have been observed at room temperature in either single crystalline or polycrystalline samples[26,27]. Furthermore, possible effects of ion migration were not taken into account. Even though recent work conducted by Rohm et al.[28] showed polarized ferroelectric domains present in each grain using PFM, there was still no observation on switchable states relevant to domain changes by applied electric field[29]. Meanwhile, Strelcov et al. reported the existence of ferroelasticity controlled by external stresses in single and polycrystalline films instead of ferroelectricity, and also showed switchable states under applied stress. These studies point to the possibility of exploring domain properties and their role in perovskite solar cell devices.

In this work, complementary spatial nanoscale imaging techniques, specifically PFM, Kelvin probe force microscopy (KPFM), and AFM are used to identify structural changes of (FAPbI$_3$)$_{85}$(MA-PbBr$_3$)$_{15}$ perovskite under varying illumination and bias conditions. Firstly, nanoscale ferroelastic domains are identified using PFM measurements. In-situ temperature-dependent topography measurements confirm that the domains disappear and reappear at the phase transition temperature. These domains have a strong dependency on external light and electrical bias. When exposed to light, the PFM amplitude greatly increases, thus, domain patterns are clearly visible in both amplitude and phase maps. Structural appearance of domains is also observed in 3D topography maps. Secondly, the effect of bias voltage is observed by applying the voltages to the tip during the PFM measurement. Upon applying positive bias voltages, strong domain patterns appear in both phase and amplitude maps with enhanced amplitude signal. The domains distinctly emerge structurally which is confirmed by a 3D topography map. However, upon applying negative bias voltages, domains are no longer observed structurally, rather, migration of organic cations dominates and causes damage to the surface. Peak broadening analysis from in-situ XRD measurements reveals increase of structural disorder under the applied external stimuli, i.e., the light and electrical bias which is linked to the appearance of the domain. Lastly, we show evidence and propose a mechanism describing how the observed structural changes can influence charge carrier transport and the I–V hysteresis in actual solar cell devices.

## Results

**Scanning probe based characterization.** Figure 1a schematically illustrates the experimental set-up for the PFM measurements. The procedure of sample preparation is described in the experimental methods section. All experiments are carried out under ambient conditions and low humidity conditions below 20%. Our results are consistent over several repetitive scans so we expect that there is no degradation during the measurement. The structure consists of FTO/c-TiO$_2$/m-TiO$_2$/(FAPbI$_3$)$_{0.85}$ (MAPbBr$_3$)$_{0.15}$. Compositional engineering of a mixed cation/metal halide perovskite in the form of (FAPbI$_3$)$_{0.85}$(MAPbBr$_3$)$_{0.15}$ has been shown to be one of the most effective ways to realize record-efficiency perovskite solar cells[30].

Figure 1b shows an out-of-plane PFM amplitude map under dark and light conditions. The light was turned on half way into the scan (slow scan axis is from bottom-to-top). In dark, the amplitude map shows a rather low response with average value in the range of 900–1000 arbitrary units (a.u.). As soon as the light is turned on half way through the measurement, the signal is greatly enlarged up to 2100 a.u. Further, under illumination, high-magnification PFM amplitude and phase maps (Fig. 1c, d) resolve the presence of stripe domains for the grains in the polycrystalline sample. We notice that the stripes do not exist for every grain. One reason can be that solution processed polycrystalline perovskites cannot have perfect and homogenous composition all over the film. Especially, halide segregation is often observed in solution processed mixed halide perovskites[31]. Previous PFM results also do not show domain patterns for every grain, even in single cation-based perovskites[29,32] Another noticeable phenomenon is that the domains start to emerge clearly as stripes under illumination only (Fig. 1c, d). However, under dark conditions these domains fade away (Fig. 1d, e). These results imply that the light illumination can induce significant increase in piezoresponse and is possibly associated with the large density of domains and walls. It was previously reported that the effective piezoelectric coefficient increases under white light due to light-induced rearrangement of the molecular dipole moment, resulting from the looseness in hydrogen bonding of the excited perovskite frameworks[26]. The spacing between the domains ranges from about 40 to 350 nm which is consistent with the domains observed in other reports[16,32,33].

Light illumination on halide perovskites provokes various phenomena such as phase segregation[34], photo-induced lattice expansion[35], and ion migration[36]. Below we consider the possibility that these properties are responsible for the observed emergence of domains. We have previously observed that ion migration gradually proceeds over several minutes and grain boundaries act as facile migrating channels[37]. Supplementary Fig. 1 compares PFM magnitude and CPD maps and light is turned on half way from the measurement. It is apparent that CPD contrast at the grain boundaries becomes visibly brighter under illumination whereas grain boundaries in the amplitude map do not change under illumination. Also, it is apparent from Supplementary Fig. 1. that the increase in amplitude is abrupt instead of a gradually increase over time. To further examine the effects on ion migration on the PFM signal, we examined the

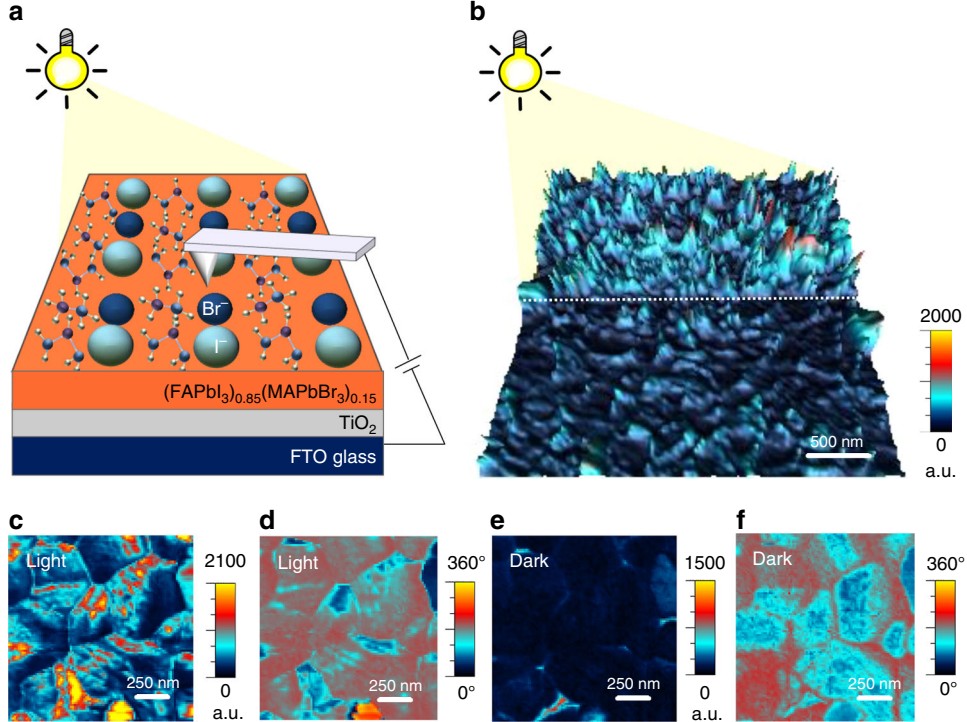

**Fig. 1** Light-induced piezoresponse signal enhancement and stripe domains. **a** Schematic illustration of photo-assisted scanning probe microscopy (SPM) with a perovskite film on compact-TiO$_2$/mesoporous-TiO$_2$/FTO test structure under illumination. **b** 3D piezoresponse amplitude map of (out-of-plane) with and without illumination. **c** Out-of-plane amplitude map in PFM under light condition, and **d** the corresponding out of plane phase map. **e** Out-of-plane amplitude map in PFM in dark condition, and **f** the corresponding out of plane phase map

PFM signal before and after the light illumination. Supplementary Fig. 1. shows that PFM signals return to the initial dark values as soon as light illumination is turned off. These results imply that the ion migration does not have direct effect on the above observed PFM results. However, we cannot completely rule out the effects of ion migration which also occurs within the film and change the octahedral distortion which affects the stress within the film[38].

We performed photoluminescence (PL) spectral measurements over several minutes to examine the effect of phase segregation due to the light illumination, i.e., bromide rich phase and iodide rich phase. It is clear that peak splitting does not occur over time under the illumination as shown in Supplementary Fig. 1.

To elucidate the origin of the domain patterns, we performed in-situ temperature-dependent topography measurements as shown in Supplementary Fig. 2. It is expected that the domain patterns will disappear upon annealing above ~57 °C, as the perovskite film undergoes a phase transition from tetragonal to cubic, releasing internal strain. We expect that phase transitions in different crystal grains have different transition temperatures due to compositional non-uniformity. Therefore, not every grain should have the phase transition or the same phase transition temperature. We carefully ramped up the temperature (2 °C min⁻¹) slowly so that there was no significant thermal stress generated. It was not possible to perform AFM contact-mode measurements at high temperatures as the surface of the sample was then easily damaged due to serious ion migration when the tip is directly in contact with the sample surface. Therefore, only non-contact mode topography measurements could be performed. 3D topography images as a function of temperature are shown in Supplementary Fig. 2. Supplementary Fig. 2 also shows line profiles of the corresponding coloured regions in the 3D topography image. The domain patterns in Supplementary Fig. 2 gradually diminish as the temperature increases. Once the temperature reaches 72 °C, the domain patterns almost vanish.

We then slowly decrease the temperature to room temperature and the domain patterns reappear in some of the grains. After the phase transition, corrugated structures of the surface re-appear in the same region although not with completely the same structures. At the same time, KPFM measurements were carried out as a function of temperature as presented in Supplementary Fig. 3. The stripe domain patterns are strongly present at room temperature while such patterns were unambiguously annihilated above 72 °C which are also apparent in the line profile data. These results are in good agreement with the recently reported TEM results on MAPbI$_3$ perovskite[33]. Twin domains in tetragonal MAPbI$_3$ have been directly observed using TEM and a reversible process, that is, the appearance and reappearance of the domains, was observed at the phase transition temperatures. In fact, the spacing between the domains ranging from around 100 to 300 nm wide observed in that work, is in a similar range observed in this work. The grains in tetragonal phase regarding the cubic parent grains can have six crystallographic orientation states, which can be possible to make them a ferroelastic material according to empirical definition[39]. Therefore, we suspect that the observed domain patterns in the films are ferroelastic domains, which are generated during the cooling of the film. It is possible that a cubic to tetragonal phase transition during film processing could induce an internal strain which is released from the existence of twin domains caused by ferroelasticity[33]. Since each grain could have differences in elemental distribution and structure, especially if the sample contains multiple cations and halides, the independent phase transition occurs in individual grains, thus, there exist the distinctive patterns in each grain.

To investigate this effect further, we performed high-resolution morphological imaging of the polycrystalline thin film. Figure 2a, b show 3D topographic images of the grains with stripe domain patterns in light and dark conditions respectively. Figure 2c shows the corresponding height profiles for the topography images

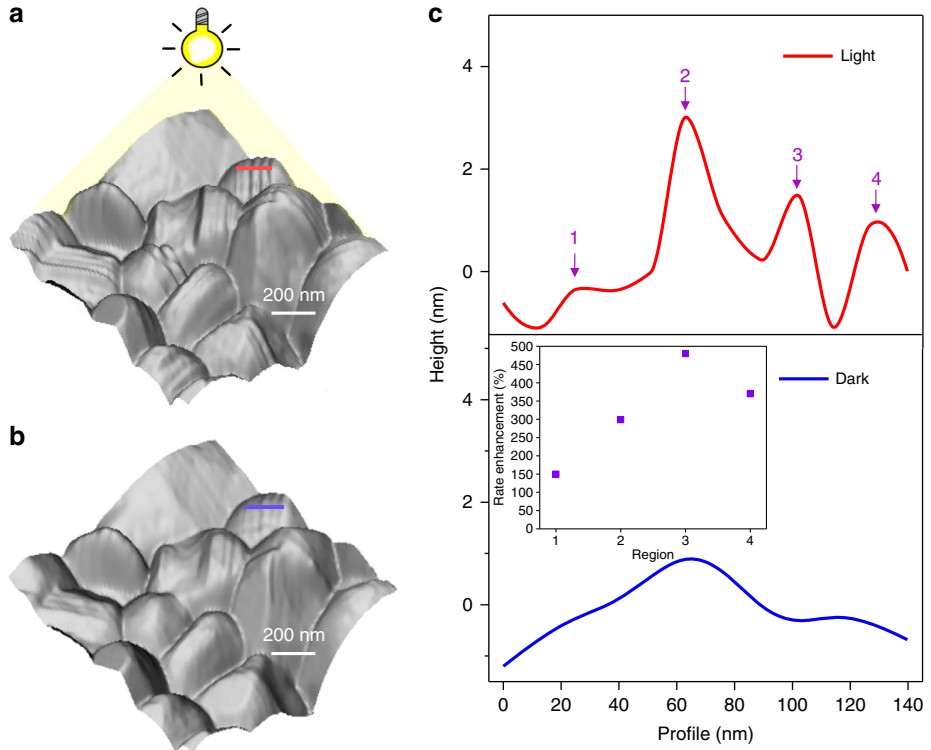

**Fig. 2** Light-induced structural surface variation of MA-FA perovskite. 3D topography **a** with light, **b** without light, and **c** corresponding line profiles for topography in under light (red) and in the dark (blue). The inset represents rate enhancement at the marked regions

measured under light (red) and in the dark (blue). The domain patterns emerge strongly under light illumination for some grains. Observation of domain patterns in the topography map is due to surface corrugations associated with matching of mechanical compatibility conditions imposed by the neighbouring ferroelastic domains. The height difference between each domain is −1 to 1 nm in the dark while it turns to −1 to 3 nm under light. It is important to note that such phenomenon does not occur in every grain. Rather, the appearance of the domain patterns is dependent on the specific grain, which could be due to the non-uniform elemental distribution between grains which also leads to variation in structural properties[31].

In order to investigate the effect of bias on the evolution of ferroelastic domains, bias-dependent PFM measurement in dark conditions were performed to exclude the effects of photo-induced ionic motion or excited charge carriers. When high bias is applied to the tip during PFM measurements, local damage to the surface of the films occurs owing to strong ion migrations and sensitively fragile film surfaces. We checked that the sample is stable towards positive bias up to +2.5 V applied to the tip, whereas degradation starts when negative bias larger than −0.5 V is applied to the tip with applied AC bias of 2 V. Structural damage induced by large (>−4 V) negative bias is irreversible in our previous report[37]. The ionic motion depending on the positive and negative biases applied to the AFM tip is illustrated in Supplementary Fig. 4[37]. The effect of applying a bias voltage whether positive or negative is apparent from the amplitude plot in Fig. 3a. It can be seen that the amplitude always increases with bias voltage. Figure 3c, d shows the representative amplitude and phase maps acquired with a DC bias voltage of +2 V while Fig. 3e, f shows the same maps when applied DC voltage equals −0.2 V. The full series of the out-of-plane amplitude and phase maps are shown in Supplementary Fig. 5. Note that the measurements are performed consecutively from 0 to +1 V, +2, 0, −0.2, −0.5, and then 0 V again. In addition to the dark

condition, bias-dependent PFM measurements were carried out under light illumination as shown in Supplementary Fig. 6. The ferroelastic domain patterns become clear in the amplitude map acquired at a DC bias of +2 V (Fig. 3c). Interestingly, the response resembles that under light illumination in the absence of a DC bias (results shown in Fig. 1b). The average amplitude signal of the map under light illumination (from Fig. 1b) is ~1200 a.u. which is close to the results when +2 V is applied as shown in Fig. 3a. These results indicate that light illumination, thus, effect of photo-generated carriers, is not solely responsible for the increase in the piezoelectric response of the film. We also observe a switching of domains (PFM phase signal) when different polarity of bias voltages is applied as shown in Fig. 3b. The profiles are extracted from marked black and red lines in Fig. 3d, f. Specifically, the profiles exhibit alternating phase contrast along the domain pattern under an applied bias of +2 V and is opposite to the profile in the case of an applied voltage of −0.2 V. This flipping of the phase contrast possibly arises from external bias-induced stress which will be further clarified using XRD measurements.

Also, the bias-dependent response under illumination was qualitatively similar to that obtained under dark conditions (see Supplementary Fig. 6 for details). Such an inversion of the phase contrast may imply existence of ferroelectricity. We thus performed ferroelectric switching experiments to examine whether the change in stripes contrast originates from ferroelectricity of the perovskite. Results are shown in Supplementary Fig. 7. Our results show that no reproducible evidence for polarization–electric field hysteresis loops are found in the experiments. Recently, polarized domain patterns have been observed using PFM in MAPbI₃, however, switching of phases has not yet been observed. Therefore, these well-aligned stripe domains in our investigation are ferroelastic, and consequently do not show electrically switchable hysteretic piezoelectric behaviour consistent with earlier studies[29,32,33]. However, the ferroelastic

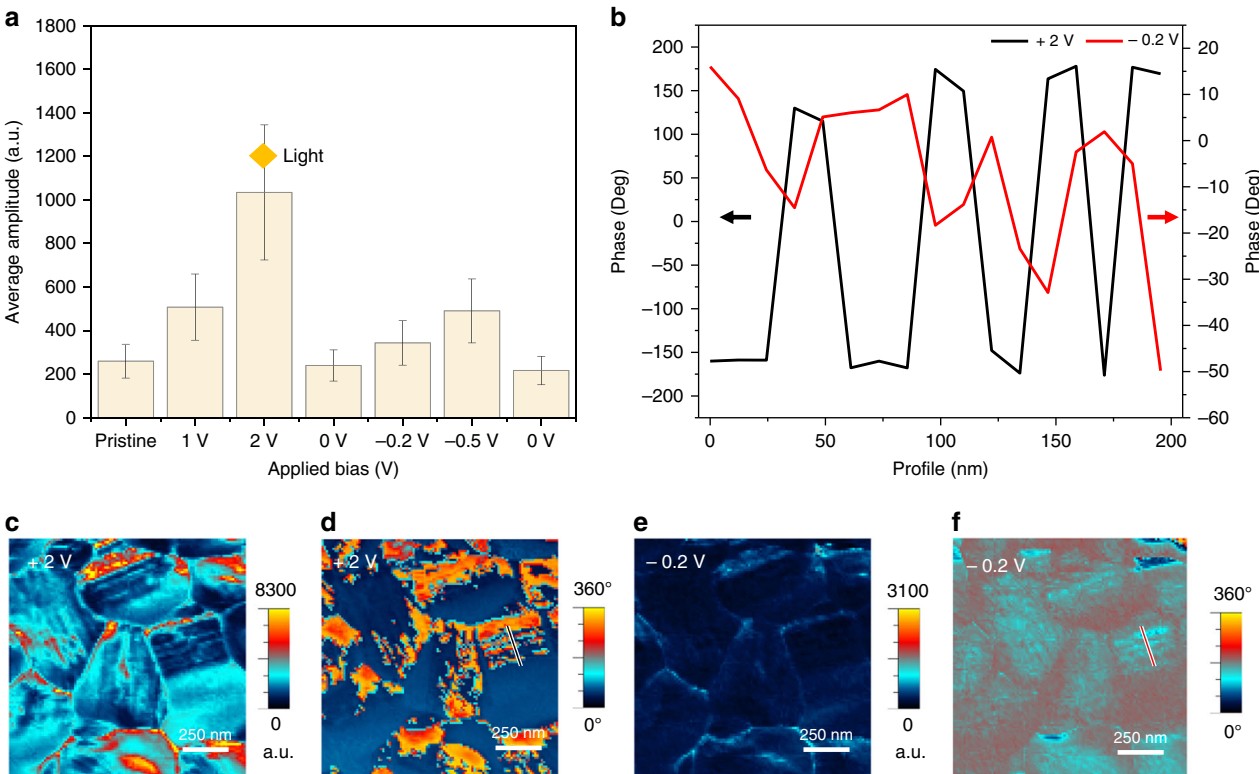

**Fig. 3** Effect of applying positive and negative bias to the film surface. **a** Graph of average amplitude signals under different bias. **b** Line profile of black line in **d**, and red line in **f**; PFM **c** out-of-plane amplitude map and **d** out-of-plane phase map under bias voltage of +2 V; **e** out-of-plane amplitude map and **f** out-of-plane phase map under bias voltage of −0.2 V

domain structure couples indirectly to an applied external stimulus, which can be in form of an optical illumination or the electrical bias. The applied external stimulus causes ion migration through the sample and can lead to generation of stress or stress gradients of sufficiently high magnitudes. These elastic gradients then in turn can influence ferroelastic domains.

We now move on to the observation of the structural variation of the film upon applying bias voltages in dark. Figure 4a–c shows representatively topographic images in contact mode after applying +7 and −7 V. It is important to note that the previous PFM measurement (Fig. 3) requires AC bias applied to the sample and surface damage occurs at much lower bias voltages. Figure 4d shows the corresponding height profiles of the black, blue and red regions in the topography images of Fig. 4a–c. As for domain patterns, it is worth noting that stripe patterns appear strongly with positive bias showing 10–30 nm of height difference which is much higher than those observed in Fig. 1. However, such patterns almost disappeared when negative bias was applied, although slight corrugated structures are still seen in Fig. 4d. We investigated structural variations of the surface when higher biases were applied but degradation at grain boundaries occurs as can be seen in Fig. 4c under −7 V voltage bias. Also, there is a new hump generated. We have previously shown that organic cations (MA$^+$ and FA$^+$) can escape to the top surface or dissociate and form such a hump, whereas, migration of halides do not damage the sample even at higher voltages[37]. As realized by the increase in the PFM amplitude signal (Fig. 3a), it would be possible that the negative bias voltages can also form the domain patterns similar to the positive voltages which is also evident that the negative voltages increase the PFM amplitude signal (Fig. 3a). However, it seems that the structural damage is readily induced by migration of organic cations under such conditions.

**XRD analysis**. To further clarify the observed structural and phase variations, we performed XRD measurements of the same test structures under similar conditions: light illumination vs dark and different levels of voltage bias. We deposited a thin layer of gold contact (5–10 nm) on top of the perovskite film for biasing the film normal to its surface. Detailed studies show that full width at half maximum (FWHM) of XRD diffraction peaks increase upon light illumination. We confirmed that there is no major damage during the measurement as the PbI$_2$ peak does not change significantly (PbI$_2$ being one of the expected degradation products reported before[40]) and return to the almost original pattern after applying the stimuli (Supplementary Fig. 8). We note however, that some extend of amorphisation of the material could be present as shown by decreased peak intensities (Supplementary Note 1).

Primarily it is clear that peaks become broader under light illumination in XRD patterns. In particular, we zoomed the peak at around 37.5° that broadened the most and found that the peak returns to the original state only after the illumination but not after the bias (Supplementary Fig. 8). Also, the peak at around 44° (Supplementary Fig. 8) does not change under the light, however, the new peak extends from 43° (300) and has a maximum at around 44°. We compared peak intensities before and after the external stimuli for the first two peaks (13.9 and 19.5°) (Supplementary Fig. 8) in order to confirm crystallinity of the sample before and after the external stimuli. Under light illumination, the peak intensities return almost to the original values. However, significant changes are observed after applying bias voltages. For positive voltages, both peaks increase and slightly decrease when 0 V is applied again. Then, the peak intensities almost do not vary upon negative voltages and remains the same when 0 V is applied again. The reduced intensities after applying voltages suggest that the sample undergoes irreversible

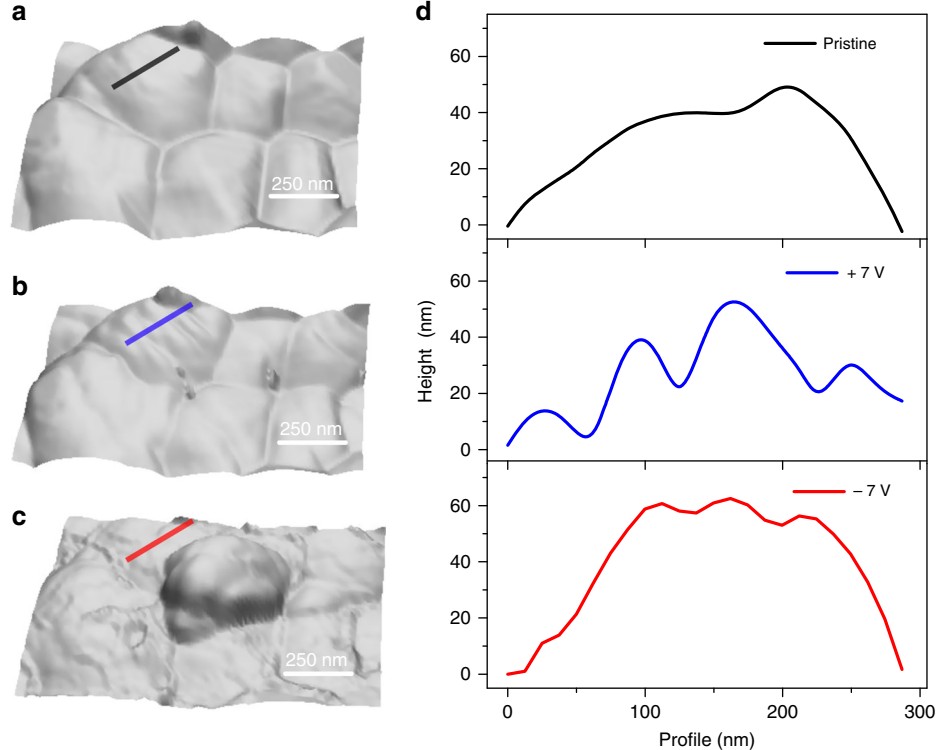

**Fig. 4** Effect of bias on structural variation of the surface. **a** 3D topography in non-contact mode in the dark, **b** after positive bias, +7 V, and **c** after negative bias −7 V. **d** Line profiles for the black, blue, and red regions in **a**–**c**

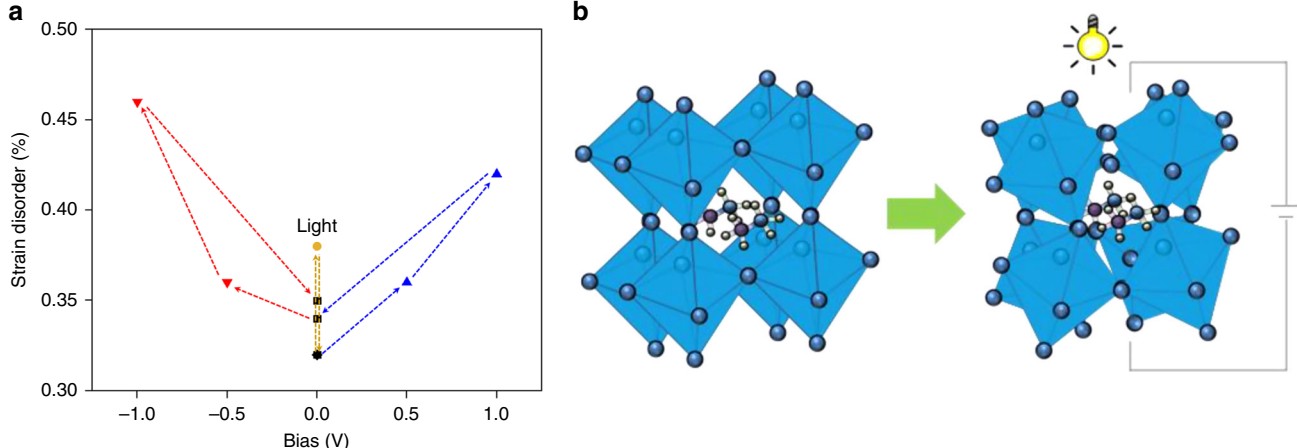

**Fig. 5** Photo-induced and bias-dependent Williamson–Hall analysis. **a** Strain disorder variations under bias and light from successive in-situ XRD measurements (see also Supplementary Figs. 8 and 9). **b** Schematic illustration of the octahedral distortion induced by bias voltage and light illumination

amorphisation to some extent (reduced crystallinity) after the bias voltages. Such reversible and irreversible variation of the peak is responsible for both reversible and irreversible chemistry and structural rearrangements, depending on the magnitude of applied bias or light intensity which can induce the observed peak broadening[41,42]. Also, the peak broadening can be originated from reversible phase segregation[34] or reversible electric field induced phase conversion[41,42].

This peak broadening can be systematically studied by Williamson–Hall (W–H) analysis in order to analyse strain disorder induced by light and bias as shown in Fig. 5a. We have only chosen the (100) and (110) planes for the W–H analysis which is explained in Supplementary Note 2. The strain variation

of light-induced metal halide solar cells ranges from 0.32 (dark) to 0.38 (light) in Supplementary Fig. 9. Under illumination, applied strain is around 20% higher than that in the dark. Such a difference in strain disorder upon illumination would suggest that it is enough to generate structural variation not only on the surface, a so-called distorted tetragonal shape. This strain variation can explain why corrugated structures of the surface are very strong under illumination, as shown in Fig. 2. Looking at the effect of positive bias shown in Supplementary Fig. 9, the increase in strain disorder is moderate: around 10% when the applied voltage is <0.5 V. However, there is a linear growth in strain when +1 V is applied. The strain is ~30% higher than that in dark condition. These results correlate with bias-dependent

PFM measurement previously discussed in Fig. 3. As expected, it is obvious that positive bias makes stripe domains become clearer with light, and also PFM signals stronger including increases in strain disorder as shown in Figs. 2 and 4. We can also see that negative bias (Supplementary Fig. 9) has a higher effect as changes can be seen even at −1 V. Consequently, cations such as organic ions give rise to a large strain rather than anions such as halide ions. Both biases play an important role in generating strains at the surface and bulk including ionic movement.

It is important to note that the effects of light and bias reported and discussed in the previous section only examined the surface of the sample using SPM. It will be useful to examine ionic motion horizontally in future work. Nevertheless, we can conclude from Fig. 5, that light and bias generate strain disorder as well as structural change on the surface. In addition, much smaller sized organic ions can create strain if we exclude light-induced halide ion movement in the dark condition with negative bias.

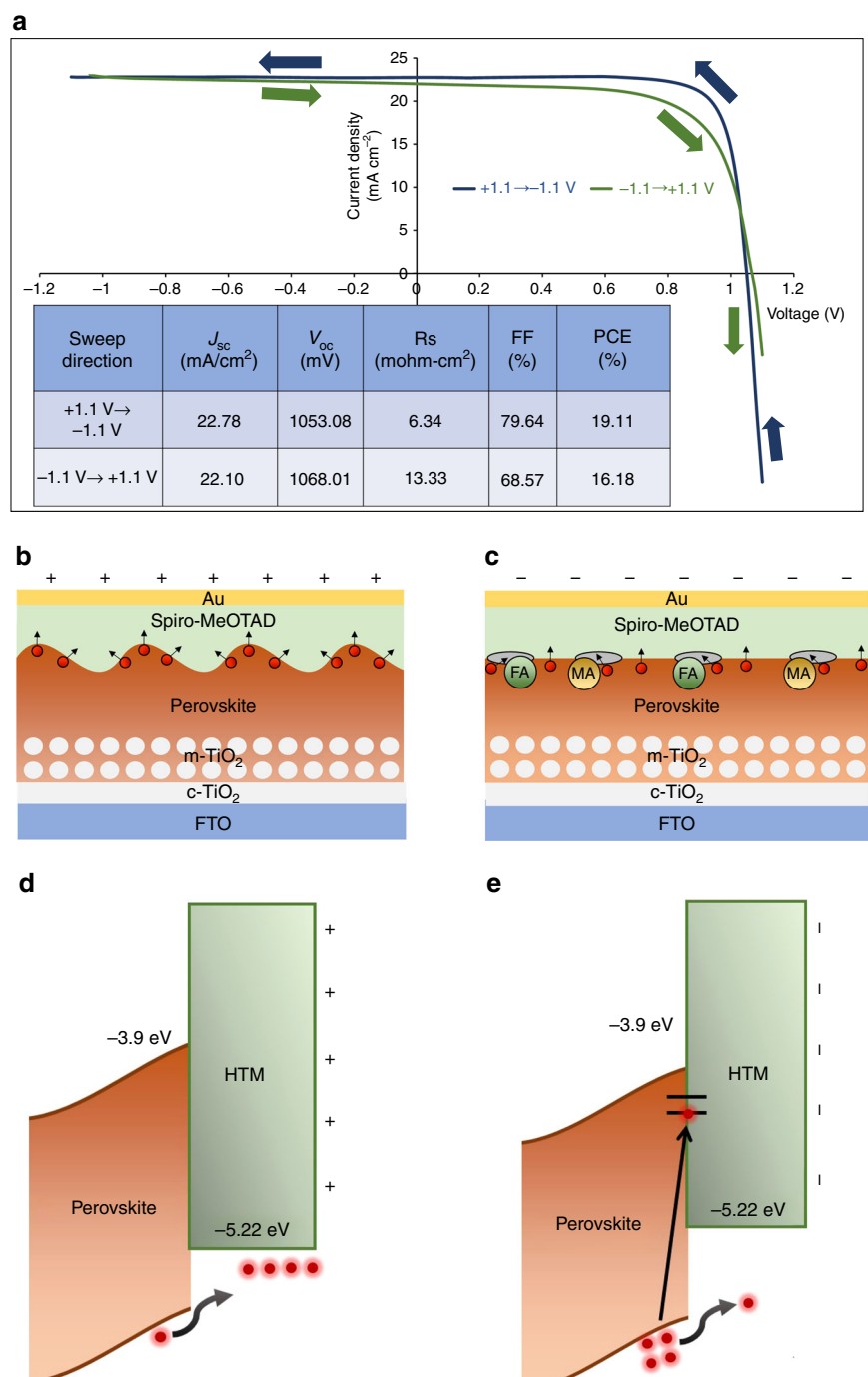

**Fig. 6** Proposed mechanism of the I–V hysteresis caused by structural variations. **a** Forward bias (+1.1 to −1.1 V) and reverse bias (−1.1 to +1.1 V) I–V curves measured at a rate of 0.1 V s$^{-1}$. Inset shows device performance parameters. Schematic illustration of perovskite surface when **b** positive bias and **c** negative bias are applied to the top surface. Corresponding band diagrams of perovskite/HTM interface when **d** positive bias and **e** negative bias is applied to the HTM

**Electrical transport measurements**. Furthermore, we investigate the impact of our previous observations on charge transport and the $I$–$V$ hysteresis of actual halide perovskite solar cell devices. Here, we propose a mechanism illustrated in Fig. 6. $J$–$V$ curves of perovskite solar cells measured with reverse (−1.1 to +1.1 V) and forward (+1.1 to −1.1 V) scans are shown in Fig. 6a. A relatively high scan rate is used (0.1 V s$^{-1}$) in order to evaluate the hysteresis effect and minimize the effect of slow ionic motion. As can be seen, there is a large performance difference. It is noticeable that FF (%) is significantly lower when measured from reverse scan due to increase in series resistance when measured from reverse bias. When $J$–$V$ curve is scanned from the positive bias (reverse scan), the domains emerge structurally which is illustrated in Fig. 6b. These corrugated surfaces will provide multi-directional pathways for photo-generated charge carriers (holes) to be transported to a hole transport material (HTM), Spiro-MeOTAD. For an opposite bias, the surface of the film is going to be modified by migration of organic cations and removes the domain patterns or it can even cause severe damage to the perovskite-HTM interface as observed in Fig. 4c and this is illustrated in Fig. 6c. Consequently, lower series resistance is expected from the forward bias scan due to an effective hole extraction.

In order to observe charge separation behaviour of the domains, KPFM measurements under the light illumination were performed. Supplementary Fig. 10 shows a topography image with domains being present. When +2 V are applied to the tip during the KPFM measurement, the contact potential difference (CPD) varies within the grains similar to what is observed in the topography map as shown in Supplementary Fig. 10. On the other hand, when −2 V is applied, no such variation within the grains is observed (Supplementary Fig. 10). Also, Supplementary Figure 10 shows CPD distributions, and median CPD is 59 mV higher when +2 V is applied to the tip compared to that of −2 V. This implies that the charge separation is more effective when +2 V is applied to the tip. A relevant energy band diagram of the perovskite-HTM interface when the positive bias and negative bias are applied to the top surface is illustrated in Fig. 6d, e respectively. Holes are actively transferred to the HTM from the perovskite layer for the positive bias whereas holes are piled up or can be trapped at the interfacial traps generated by organic cations for negative bias. Of course, there are also other factors such as ion migration, which has been convincingly linked to the origin of the $I$–$V$ hysteresis that adjusts band alignment at the interfaces[18–21]. However, our results clearly imply that the observed nanoscale structural variations can also cause imbalanced charge transport depending on sweeping bias polarity. There is a chance that surface modification occurs after deposition of other layers in solar cell devices, so the domains may not be the same as we have observed here. However, based on our XRD results, the strain under light and bias will occur within the film bulk.

## Discussion

Our findings clarify stripe pattern formation in halide perovskite grains and shed light onto stripe domain properties under changing light, bias and temperature conditions. Under illumination, corrugated structures on film surfaces were observed to be very strong, which also raised the PFM amplitude signal. Interestingly, the structure of the films when positive bias was applied shows similar results to the case under illumination, while negative bias removes uneven structures on the surface. Williams–Hall plot of XRD results show a pronounced correlation with strain disorder induced by light and bias. We conclude that the observed striation domains form in some grains that have

corrugated structure, and their properties can be controlled, leading to strain disorder with changed external conditions such as light, bias, and temperature. Moreover, we propose a mechanism that such structural variation can cause $I$–$V$ hysteresis. These results provide further understanding of the effects of nanoscale structural variations in solar cell operation and for improving the design of halide perovskite photovoltaic devices.

## Methods

**Solar cell fabrication**. The spray pyrolysis deposition method was used for a dense TiO$_2$ blocking layer of a 60 nm thickness (Compact-TiO$_2$) to deposit it onto a F-doped SnO$_2$ (FTO, Pilkington, TEC8) substrate. Here, a $20 \times 10^{-3}$ M titanium diisopropoxidebis (acetylacetonate) solution (Sigma Adrich) at 450 °C was also used to avoid direct contact between hole-conducting layer and FTO substrate. Approximately 80 nm think mesoporous TiO$_2$ (m-TiO$_2$, Dyesol 30 NR-D) that has average roughly 50 nm nanoparticles, anatase phase, films were dissolved and coated onto the Compact-TiO$_2$/FTO substrate by spin coating method with 5000 rpm for 10 s. The films were annealed at 100 °C for 10 min and calcined at 500 °C for 1 h in air for removal of the organic matter. Fabrication methods of the metal halide perovskite solutions are as follows: MAI (CH$_3$NH$_3$I) and FAI (NH$_2$CH=NH$_2$I) were firstly synthesized with reacting 30 mL hydroiodic acid (57% in water, Aldrich), 27.86 mL CH$_3$NH$_2$ (40% in methanol, Junsei Chemical), and 15 g formamidine acetate (Aldrich) in a 250 mL round-bottom flask at 0 °C for 2 h with stirring. The precipitates were obtained using an evaporator at 50 °C for 1 h. The precipitated materials were dissolved in ethanol, recrystallinized with diethyl ether, and then dried at 60 °C in a vacuum oven for 24 h. As above, CH$_3$NH$_3$Br and NH$_2$CH=NH$_2$Br were also prepared with hydrobromic acid (48 wt% in water, Adrich). The desired perovskite solutions of 85FAPbI$_3$ 15MAPbI$_3$ were spin-coated onto the mesoporous-TiO$_2$/compact-TiO$_2$/FTO substrate. The films were finally dried on a hot plate at 100 °C for 20 min.

**SPM measurements: AFM, PFM, KPFM**. The PFM measurements were performed using a commercial AFM system (AIST-NT SmartSPM 1000) under ambient conditions at room temperature. Nitrogen gas was used for cleaning sample surfaces before measurements. PFM images were taken on the surface of the films after they were cleaned with nitrogen gas using Pt-coated (HQ:CSC 37/PT, μmasch, Force constant, $k = 0.8$ N m$^{-1}$) Si cantilevers that possesses a range of resonance frequencies from 30 to 55 kHz for AFM and PFM. AC bias (1500–2000 mV) was applied to the PFM tips, the lowest one among the resonance peaks was chosen. All PFM data were obtained with out-of-plane and in-plane amplitude and phase maps at the same scan. Light sources were blocked for dark conditions. Tuneable white LED light sources that are not filtered were used for illumination at an intensity of 0.3 to 0.5 W cm$^{-2}$. DC bias was applied towards the tip or sample. KPFM measurements were carried out using diamond-coated conductive probes (DCP20, force constant, $k = 48$ N m$^{-1}$) possessing a resonance frequency of 420 kHz. A heating stage with temperature controller (SRS, PTC10) was used for temperature-dependent AFM and KPFM measurements. In all PFM, AFM and KPFM measurements, the used scan rate is 1.0 Hz, and scan direction starts from bottom to top.

**XRD measurements**. The peak broadening of MA-FA perovskites was analysed using X-ray diffraction (PANalytical, Empyrean). LED white light located inside the diffractometer is used for light soaking of the sample at 0.3 kW cm$^{-2}$. The samples were connected to a controllable power supply with wire connections to apply positive and negative bias after Au deposition (EMITECH, K550X) onto the surface.

## Data availability

All data used in this manuscript are available from the authors on request.

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

## Acknowledgements

This research has been financially supported by the Australian Government through the Australian Research Council (ARC). Responsibility for the views, information or advice expressed herein is not accepted by the Australian Government. We acknowledge the facilities, and the scientific and technical assistance of the Electron Microscope Unit (EMU), the University of New South Wales (UNSW).

## Author contributions

D.K. and J.S.Y. conducted the PFM, AFM and KPFM measurements, XRD analysis, analysed the data and wrote the paper. P.S. analysed the data. D.S.L. and J.K. fabricated the films with solution processes. J.K. measured J–V hysteresis curve. A.M.S. performed PL measurements and analysed the data and wrote the paper. S.H., M.A.G. and A.W.Y.H. supervised the project and co-wrote the paper. J.S. conceived the idea, and led the project, analysed the data, and co-wrote the paper. All authors contributed to the manuscript.

## Additional information

**Competing interests:** The authors declare no competing interests.

