## [Peer Review File · Nature Communications]

Reviewers' comments:

Reviewer #1 (Remarks to the Author):

The manuscript by Kim et al. presents scanning probe microscopy work on metal halide perovskite thin films to investigate the structural variations under various illumination and bias conditions. The major finding is that the stripe domains observed in some grains are ferroelastic and can be modulated by light and electric bias. Combined with XRD measurements, it is shown that the ferroelastic domains are associated with the strain induced by light or bias. Based on these results, the authors proposed a mechanism of the I-V hysteresis caused by structural variations. The work is certainly of interest to the perovskite solar cell community. As it stands now, however, the paper is not suitable for publications in Nature Communications for a number of reasons detailed below.

1) Ferroelastic stripe domains have been previously observed and thoroughly studied in Ref. 29, where the data quality is generally better there than that in the present work. While the result here adds some new understandings, it is rather incremental than original. Unless the authors can fully address the following questions or provide evidence that the present work is truly novel, I would recommend the re-submission to Sci. Rep. or other specialized journals.

2) Light illumination can induce both ionic motion and free carrier conduction. In this work, the authors illuminated the sample with a few hundred mW/cm² white light (several Sun), which certainly creates large amount of conduction electrons. Their effect on PFM measurement (a few volts in AC and DC bias) needs to be clarified. For instance, illumination with below and above the gap with different power density would be preferred to elucidate the light-induced structural variations.

3) The reason why stripe domains only show up in some random grains under external stimuli (light and bias) is not yet clear. In fact, the difference between images in dark and under illumination (Fig. 2 of this paper) is not striking at all. One can already see the presence of stripes, which are just moderately enhanced by light. To investigate the origin of ferroelastic domain pattern, the underlying mechanism of the relations between grains, corrugated features, and light/bias stimuli should be clearly addressed.

4) The authors performed P-E loop measurement at one point within a grain and observed no butterfly curves in both amplitude and phase. This, however, is not sufficient to exclude the possibility of the existence of ferroelectric domains, since the grain does not seem to have corrugated patterns which is claimed to be related to domain induction (Figure S7). In a previous study, stripe ferroelectric domains were observed in MAPbI₃ (Ref. 28 in this paper). Thus, more experiments and analyses are required to justify the statement.

Finally, the authors need to carefully proofread the manuscript for simple typos. As a few examples, on Page 3, "complimentary" should be "complementary". On page 14, "cause" is missing in "it can even severe damage to...".

Reviewer #2 (Remarks to the Author):

In this paper, Kim & Yun et al. report on the investigation of structural variations in perovskite films at the nanoscale under excitation. Based on different techniques, the authors show the formation of ferroelastic domains that are induced by light and/or bias stimulation. Interestingly, these effects

occur within seconds and can eventually be related to the fast efficiency decay observed in PV devices. This work is expected to give insight into the origin of structural modifications in perovskite films induced by excitation, which represents one of the most active debates in the field and thus is a very timely topic. Therefore, I feel this manuscript is suitable for the broad readership of Nature Communications. However, prior to its acceptance, the authors should address some major concerns accordingly.

1. How uniform spatially is the illumination? It is possible that local striations appear on samples when coupling in a laser/lamp through a microscope tip. Authors should image their illumination onto the camera to make sure they are not imprinting a striation artefact.
2. How much are the domains/effects an artefact of the way it is scanned? Authors should provide details about scan rate, scan direction, etc.
3. It is well known that the atmosphere in which the sample is placed dramatically affects the properties of ABX₃ perovskites under illumination (DOI: 10.1021/acs.jpcllett.5b00785, DOI: 10.1126/sciadv.1600534, DOI: 10.1021/acs.jpcllett.8b01830, etc). In fact, most of the measurements have been performed in ambient conditions, therefore authors should describe in detail whether the atmosphere could boost or mitigate their observations.
4. Authors performed PL measurements in bulk perovskite to show there is no phase segregation. Nevertheless, it is still possible for nanoscale segregation that doesn't manifest itself in the macro.
5. How do the authors avoid the injection of charge carriers with bias?
6. Could authors comment on the fact that similar (in magnitude and dynamics) stresses are produced by applying bias or light? It would be needed to provide information regarding the magnitudes of the local electric fields induced in the film by the different excitation sources and why they result in similar stresses (for example see Fig 3A, 2V vs light).
7. The application of negative bias gives rise to structural damage induced by migration of organic cations. Is the process reversible and/or accumulative? Several cycles in which positive and negative bias are sequentially applied would provide insight here.
8. The conclusions have important implications when correlating them with solar cells. In this regard, the authors should comment on any evidence for the existence of the stripes in the complete stack –it is well known that after the deposition of the different layers, solvent and interface effects can modify the surface of the perovskite film-. Also, if the application of negative bias results in irreversible damage in the perovskite film, the effect should –see above- be accumulative, can the authors provide subsequent JV curve measurements and comment on that? In addition, does the proposed mechanism explain the fast or/and slow term decay in efficiency when tracking the maximum power point (DOI: <https://doi.org/10.1016/j.joule.2018.05.005>)?

Other comments are:

- In line 124, authors claim that the increase in amplitude is abrupt instead of a gradual increase over time. As far as I understand, there is no timescale in that panel.
- What is the light intensity locally? Arbitrary units are not very informative.
- CPD is not defined.
- Lines 203 and 205, I think they refer to Figures S5 and S6 instead.

Reviewer #3 (Remarks to the Author):

The major claim of the paper is that ferroelastic domains form in perovskites upon illumination and/or under applied electrical bias. The authors observe from several microscopy techniques the formation of corrugated features in the perovskite materials under illumination and/or electrical bias. The authors then use X-ray diffraction to observe changes in the structure of the materials as a function of electrical bias and illumination.

The results are interesting and (as far as this reviewer is aware) quite novel. If the analysis of the

results and data are correct, the work could be a rather influential paper. However, there are substantial questions that remain unanswered from this work. As well, it is not clear that the author's interpretation of the data is either correct or complete.

While it is clear from the microscopy that morphological changes occur when illumination and/or bias are present. It is not clear that this is due to reversible processes. While the lack of reversibility is true for both the microscopy and the XRD, I will focus primarily on the XRD since it more specifically reports on both the chemical composition and structure.

No evidence in the article is reported on the reversibility of the X-ray diffraction vs voltage/illumination. The data suggests that these perturbations (light and bias) are causing irreversible chemistry and structural rearrangements to occur. When a bias is applied, a visibly broadened peak appears at 38 degrees and a new peak appears at 45 degrees. Both peaks are rather broad suggesting limited crystallinity of the resulting material. As well, a baseline shift and overall decrease in signal intensity is also observed with applied bias; this is rather characteristic of the formation of amorphous material. This indicates that the perturbations are damaging the samples. It is presently unclear and the authors must convincingly demonstrate, that their data relates to their samples and not the damaged material. Some of the XRD data is difficult to completely assess in Figure S8. I suggest that the authors re-plot Figure S8 so that the data is clearer. Perhaps they can overlay OV on different potentials in diffractograms or alternatively zoom in on the peaks that they are analyzing.

I do not believe that the authors can effectively use Williamson-Hall analysis on their data. The lines that they fit in figure S9 are very loosely correlated with the data that they have. The fit lines are far from unique solutions and cannot with any reliability report on strain in the materials. Part of the problem is related to the number of data points that the authors include, but mostly this is related to the lack of precision in the data. Without clear correlations in the data, the Williamson-Hall analysis and conclusions related to it need to be removed from the manuscript.

Overall, I don't think that the authors can rule out phase segregation or transformation from crystalline to amorphous phases. The changes in morphology could be superficial and merely represent the upper layers of the material, while the majority of the bulk remains unchanged. As well, while the authors do not observe substantial changes at the grain boundaries, it is not necessarily so that phase segregation happens at the grain boundaries. XRD shows the formation of new peaks upon illumination and with applied bias. The XRD also shows a shift in the baseline of the pattern and general decrease in signal intensity of the peaks, which looks like the formation of amorphous material.

Temperature dependent AFM looks irreversible. I don't think that the authors are looking at the same material after they go through the temperature cycle. One way that the authors could more convincingly show reproducibility would be in Figure 1 b, demonstrating that the amplitude returns to the same level after illumination is turned off.

There are several more minor points below:

The authors need to give more details about their light source for illumination experiments. What type of lamp is it? What power density does it produce at the sample location? Was the light filtered? If so, how? What was the approximate spectral distribution? To appropriately assess the amount of energy that the perovskites receive is important for interpreting the possible physical and chemical processes that occur and the conclusions that are drawn.

Perovskites are usually very hygroscopic. What have the authors done to ensure that they do not have

water in or on their samples when performing microscopy measurements? The presence of water is known to accelerate degradation pathways under illumination (and possibly under electrical bias as well).

There are too many data points for the author's photoluminescence measurements vs time. It is difficult to see what is happening to the signal as a function of time. The authors either need to show fewer spectra or plot integrated PL intensity as a function of time.

Starting on page 3, last paragraph: The authors should label the section results and discussion. The discussion in the article is disappointingly short, one paragraph on page 15. Considering the amount of experiments that have been performed, surprisingly few points are discussed. It sounds more like the conclusion than a complete discussion, which in part is true considering that significant discussion occurs in the preceding section.

Reviewer #4 (Remarks to the Author):

The manuscript provided fundamental understanding of underlying mechanisms related to light and bias induced effects at the nanoscale in hybrid perovskite materials. Particularly, structural variations of the perovskites induced by light and bias are systematically investigated using scanning probe microscopy techniques. It was reported that periodically striped ferroelastic domains, and strain disorder were induced by these applied external stimuli. The structural emergence of domains can affect the transfer pathways for the carriers in general. The findings point to potential origins of I-V hysteresis in halide perovskite solar cells and provide fresh microscopic insight into external stimuli-induced structural disorder governing the superior performance of the metal halide perovskites solar cells. Overall, it is an interesting study regarding the ferroelastic, strain property and carrier behavior from the microscopic point of view. However, some important issues still need to be addressed before it could be further considered. The comments are listed below.

1. At present, the meso-scale expression of the ferroelastic and ferroelectric properties in hybrid perovskite materials is the formation of domains. But ferroelasticity is a structural transformation caused by force and ferroelectricity is a synergistic structural transformation caused by electrical polarization. In terms of physical nature, it's different and extremely easy to confuse between the two concepts. The author defines the light-induced domains as ferroelastic domains while these domains still be regarded as ferroelastic domains in the bias-induced part. This requires further discussion related to the physical origin of domain formation under different stimulus conditions to distinguish them. In addition, we see that the authors established the AFM characterization quite well, and therefore is it possible to dig out the mathematical relationship (linear and non-linear) of the AFM signal from the physical point of view?

2. The author attributed the light-induced ferroelastic domains to light-driven molecular reordering in line 112 and further revealed that the origin of the domain patterns is related to a cubic to tetragonal phase transition during film processing in line 159. We agree that the existence of the ferroelastic properties of MAPbI₃ is due to the asymmetrical tetragonal phase structure induced MA molecule, however currently, it is generally accepted that the perovskite of the (FAPbI₃)_{0.85}(MAPbBr₃)_{0.15} composition system is the cubic phase structure. Therefore, we would like to know whether there are also light- and bias-induced domains in pure FAPbI₃ or FA dominating materials systems in order to verify the role of MA molecules.

3. Further, the author claimed that the appearance of the domain patterns is dependent on the specific grain due to the non-uniform elemental distribution in line 174. Whether the grains with domain are all tetragonal structures? The composition is rich in MA or even pure MA? Probably TEM data and micro-zone PL could help to prove this point.

4. It is known that MAPbI₃ has phase transition from tetragonal to cubic phase near 57 °C. However,

the similar transition occurs in the mixed cation perovskite at about 72 °C is not reported. Therefore, it requires the DSC (differential scanning calorimetry) data of the $(\text{FAPbI}_3)_{0.85}(\text{MAPbBr}_3)_{0.15}$ composition. If such phase transitions occur only in certain grains rather than in the entire polycrystalline film, the question is whether phase transitions in different crystal grains have different transition temperatures due to compositional non-uniformity. I think that the author should make the point clear in the text whether this phase change is partial or integral.

5. Although various phenomena (phase segregation, photo-induced lattice expansion, and ion migration) under illumination will affect the experimental results to some extent, we believe that it is reasonable to associate the light-induced domain with the ferroelastic phenomenon. However, the photoluminescence (PL) spectral measurements over several minutes in Fig S1B cannot support the absence of ion migration. The authors may need to fit the data to provide the peak position and FWHM changes over time.

6. We know that AFM (including KPFM, PFM) is a very powerful surface characterization technique. Meanwhile, we appreciated the author by providing many exciting data to prove the domain structure changes. However, regarding the entire manuscript, most analysis the author provided are related to the film surface. Only XRD data has been presented for the structural characterization of the film bulk. However, I think the XRD data is not sufficient to support the structural variations that caused by the light- or bias-induced domain. This is because that the influence of ion migration cannot be ignored in this work, particularly the migration of halide ions I and Br under light and bias, despite that the larger Pb ions are difficult to migrate. Thus, it is recommended that authors could provide more data to exclude the interference of the ions migration.

7. There are two errors about the perovskite composition in Figure 1A and the definition associated with J-V scan direction in line 302.

8. The author proposed a mechanism that such structural variation under light and bias can cause I-V hysteresis. However, it was documented that many high efficiency devices exhibit almost no hysteresis by tuning perovskite materials, device architecture, or the transport materials. Therefore, the significance of the current results still needs to be further explored.

We would like to express our gratitude to all the reviewers for their valuable comments on our manuscript entitled “Light- and Bias-Induced Structural Variations in Metal Halide Perovskites”. Herein, we have addressed all the comments (printed in black), and provide a point-by-point response (our response is printed in red). We are convinced that our work is highly relevant to the research community in the field of perovskite solar cells and makes a contribution that can change the way we think about these important materials.

Reviewer #1 (Remarks to the Author):

The manuscript by Kim et al. presents scanning probe microscopy work on metal halide perovskite thin films to investigate the structural variations under various illumination and bias conditions. The major finding is that the stripe domains observed in some grains are ferroelastic and can be modulated by light and electric bias. Combined with XRD measurements, it is shown that the ferroelastic domains are associated with the strain induced by light or bias. Based on these results, the authors proposed a mechanism of the I-V hysteresis caused by structural variations. The work is certainly of interest to the perovskite solar cell community.

As it stands now, however, the paper is not suitable for publications in Nature Communications for a number of reasons detailed below.

Response: We thank the reviewer for the comments on our work. We have addressed all comments in detail below.

1) Ferroelastic stripe domains have been previously observed and thoroughly studied in Ref. 29, where the data quality is generally better there than that in the present work. While the result here adds some new understandings, it is rather incremental than original. Unless the authors can fully address the following questions or provide evidence that the present work is truly novel, I would recommend the re-submission to Sci. Rep. or other specialized journals.

Response: We understand that the reviewer is pointing out this paper [Ref 29: Sci. Adv. 3, e1602165, (2017)] and we also agree with the good quality of the presented data. In Ref. 29, they focused on investigating “ferroelasticity” in both single and polycrystalline MAPbI₃ thin films. They applied stress to the film, whereas we investigate effects of light and bias that are essential for solar cell applications. Thus, we think that our work is novel as we focus on the “true” solar cell related properties that have not been revealed in Ref. 29. Our work exhibits light and bias induced structural variation and properties of different types of domains that exist in mixed halide perovskite solar cell thin films. We show that the ferroelastic stripe domain structure is strongly influenced by light or bias, which is not discussed in ref. 29. Also, we have suggested a model how the light illumination and bias can influence photovoltaic devices, this will potentially influence not only their performance but also their stability.

2) Light illumination can induce both ionic motion and free carrier conduction. In this work, the authors illuminated the sample with a few hundred mW/cm² white light (several Sun), which certainly creates large amount of conduction electrons. Their effect on PFM measurement (a few volts in AC and DC bias) needs to be clarified. For instance, illumination with below and above the gap with different power density would be preferred to elucidate the light-induced structural variations.

Response: We certainly agree that carriers are generated under the illumination. As suggested by the reviewer, we studied the effect of above and below bandgap light and its intensity on the PFM measurements. In this experiment, we applied wavelength of 500nm (above bandgap) and 810 nm (below bandgap) and plotted the average PFM amplitude values as per intensity as shown in Figure R1. According to our PL spectra measurement, PL intensity peak is located between 790-800nm (please see Figure R17) which indicates the bandgap in our material. Please note that 810 nm is the absolute maximum wavelength that we can apply in our measurement setup. For the wavelength of 500 nm (above bandgap), the average PFM amplitude values linearly increase as the intensity becomes stronger. It is also noticeable that the domain patterns in both phase and amplitude maps are becoming clearer as the intensity becomes stronger (please see Figure R2). For the wavelength of 810 nm (below bandgap), the average PFM amplitude values only slightly increases even though the intensity is much stronger (up to 25 W/cm²). Consequently, the domains are not clearly visible in the amplitude and phase maps with 810 nm wavelength. These results imply that the generated carriers

have great influence on the PFM measurements. We explained that the strain increases upon the light and bias (Fig. 5) so the generation of free electrons can be also linked to the strain. The exact mechanism how the generation of free electrons are related to the strain is not known at this stage. It was previously shown that the free electrons can change the charge state of iodide ions, thus, distortion of the structural octahedra could be also changed. [Nature Mat., 17(5), 445, (2018)]

Figure R1. Laser dependent PFM amplitude with different power density

Figure R2. Out-of-plane PFM amplitude and phase maps under dark and 500 nm wavelength

The reason why stripe domains only show up in some random grains under external stimuli (light and bias) is not yet clear.

Response: We notice that the stripes do not exist for every grain. One reason can be that solution processed polycrystalline perovskites cannot have perfect and homogenous composition all over the film. Especially, halide segregation is often observed in solution processed mixed halide perovskites. [J. Am. Chem. Soc., 138, 15821 (2016)] Previous PFM results also do not show domain patterns for every grain, even in single cation-based perovskites [J. Phys. Chem. C, 120, 5724, (2016); Sci. Adv., 3, e1602165 (2017)].

We have added the above as additional discussion on page 5 of the revised manuscript.

In fact, the difference between images in dark and under illumination (Fig. 2 of this paper) is not striking at all. One can already see the presence of stripes, which are just moderately enhanced by light.

Response: As far as we are aware there is no report on enhancement of the stripe pattern induced by light illumination. We rather think that in our case the enhancement of topographically corrugated features is very strong, being enhanced by 200-300% by light illumination (Figure 2) compared to dark. Another clear image of the stripe formation is shown in Figure R3.

We have now added inset (y-axis presents rate enhancement (percentage)) in order to show clearer increase in Fig. 2.

Figure R3. Topographic images under dark and light illumination

To investigate the origin of ferroelastic domain pattern, the underlying mechanism of the relations between grains, corrugated features, and light/bias stimuli should be clearly addressed.

Response: The origin of the ferroelastic domain patterns has been reported by I. M. Hermes et al. [J. Phys. Chem. C, 120, 5724, (2016)] and by E. Strelcov et al. [Sci. Adv., 3, e1602165 (2017)]. These stripe domains are confined to individual grains because there is a uniform crystal structure within one grain. Such ferroelastic stripe domains are formed to compensate internal strain during phase transition from cubic to tetragonal. [J. Phys. Chem. C, 120, 5724, (2016); Sci. Adv., 3, e1602165 (2017)] The corrugated features are also arising from local distortion due to the ion migration [Nat. Commun., 8, 15152, (2017)] and phase segregation [Nano Lett., 16, 1485, (2016); ACS Energy Lett., 3, 204, (2018)]. This ferroelastic domain structure couples indirectly to an applied external stimulus, which can be in the form of an optical illumination or the electrical bias. The applied external stimulus causes ion migration through the sample and can lead to generation of stress or stress gradients of sufficiently high magnitudes. These elastic gradients then in turn can influence ferroelastic domains. The stimuli such as light and bias promote ferroelastic domain structures as we report in our manuscript.

To address the point raised by the reviewer, we explicitly note the following:

“These well-aligned stripe domains in our investigation are ferroelastic, and consequently do not show electrically-switchable hysteretic piezo-electric behaviour consistent with earlier studies [J. Phys. Chem. C, 120, 5724, (2016); Nat. Commun., 8, 14547, (2017); Sci. Adv., 3, e1602165, (2017)]. However, the ferroelastic domain structure couples indirectly to an applied external stimulus, which can be in form of an optical illumination or the electrical bias. The applied external stimulus causes ion migration through the sample and can lead to generation of stress or stress gradients of sufficiently high magnitudes. These elastic gradients then in turn can influence ferroelastic domains.”

on page 9-10 of the revised manuscript.

4) The authors performed P-E loop measurement at one point within a grain and observed no butterfly curves in both amplitude and phase. This, however, is not sufficient to exclude the possibility of the existence of ferroelectric domains, since the grain does not seem to have corrugated patterns which is claimed to be related to domain induction (Figure S7). In a previous study, stripe ferroelectric domains were observed in MAPbI₃ (Ref. 28 in this paper). Thus, more experiments and analyses are required to justify the statement.

Response: To further investigate whether these *corrugated grains* are ferroelectric, several additional P-E loop measurements are performed under dark condition to rule out photo-generated effects as can be seen in Figure R4. Although measurements are performed for several times, the sample shows non-ferroelectric behaviour in below figure R4. For this reason, we conclude that striation grains do not have ferroelectricity.

Furthermore, we note that there was some confusion in earlier published work, e.g. these *well-aligned* stripe domains were referred to as ferroelectric in some publications [Energy Environ. Sci., 10, 950, (2017); PNAS 114, E5504, (2017); J. Phys. Chem. Lett., 6, 1155, (2015)], while being referred to as ferroelastic in other publications [J. Phys. Chem. C, 120, 5724 (2016); Appl. Phys. Lett., 113, 072102, (2018)]. However, this question has now been settled unambiguously through a range of complementary experimental data/techniques showing that these well-aligned stripe domains are indeed ferroelastic [Sci. Adv., 3, e1602165 (2017); Nat. Commun., 8, 14547 (2017); J. Phys. Chem. C, 120, 5724, (2016)]. These ferroelastic stripe domains arise because of the coexistence of different orientations of the centrosymmetric tetragonal crystal structure [Phys. Rev. B, 2, 754, (1970); Nat. Commun., 8, 14547, (2017)]. Therefore, these well-aligned stripe domains in our investigation are ferroelastic, and consequently do not show electrically-switchable hysteretic piezo-electric behaviour consistent with earlier studies [Nat. Mater., 14, 193, (2015); J. Phys. Chem. C, 120, 5724, (2016)].

Figure R4. P-E loop measurements at corrugated grains (a-e) to check further ferroelectricity of metal halide perovskites.

We have now added this figure to the revised support information.

Finally, the authors need to carefully proofread the manuscript for simple typos. As a few examples, on Page 3, "complimentary" should be "complementary". On page 14, "cause" is missing in "it can even severe damage to...".

Response: We have fixed these typos: “Complimentary → Complementary”, and “it can even severe damage to” → “it can even cause severe damage to”

Reviewer #2 (Remarks to the Author):

In this paper, Kim & Yun et al. report on the investigation of structural variations in perovskite films at the nanoscale under excitation. Based on different techniques, the authors show the formation of ferroelastic domains that are induced by light and/or bias stimulation. Interestingly, these effects occur within seconds and can eventually be related to the fast efficiency decay observed in PV devices. This work is expected to give insight into the origin of structural modifications in perovskite films induced by excitation, which represents one of the most active debates in the field and thus is a very timely topic. Therefore, I feel this manuscript is suitable for the broad readership of Nature Communications. However, prior to its acceptance, the authors should address some major concerns accordingly.

Response: We thank the reviewer for the positive comments on our work.

1. How uniform spatially is the illumination? It is possible that local striations appear on samples when coupling in a laser/lamp through a microscope tip. Authors should image their illumination onto the camera to make sure they are not imprinting a striation artefact.

Response: We have taken optical microscope images under dark and illumination as shown in Figure R5. The illumination is spatially uniform over more than $500\ \mu\text{m} \times 500\ \mu\text{m}$.

Figure R5. Optical microscope image under dark condition with AFM tip (a) under light illumination (b).

2. How much are the domains/effects an artefact of the way it is scanned? Authors should provide details about scan rate, scan direction, etc.

Response: Stripes are not dependent on scan rate or direction, which has also been confirmed by earlier studies [e.g. see *J. Phys. Chem. C*, 120, 5724, (2016)] Periodicity measured for our stripes are the same as reported by other groups, also with other characterization methods such as TEM [Nat.

Commun., 8, 14547, (2017)]. The used scan rate in our manuscript is 1.0 Hz, and scan direction starts from bottom to top in all AFM, PFM, and KPFM measurements.

We have added this information to the methods section.

3. It is well known that the atmosphere in which the sample is placed dramatically affects the properties of ABX₃ perovskites under illumination (DOI: 10.1021/acs.jpcelett.5b00785, DOI: 10.1126/sciadv.1600534, DOI: 10.1021/acs.jpcelett.8b01830, etc). In fact, most of the measurements have been performed in ambient conditions, therefore authors should describe in detail whether the atmosphere could boost or mitigate their observations.

Response: We agree with the reviewer that the atmosphere could have some impact. In our measurements, all experiments are carried out under ambient conditions and low humidity conditions below 20%. Our results are consistent over several repetitive scans so we expect that there is no degradation during the measurement. However, we do not exclude the effect of the atmosphere that affects the charge carrier and photoemission properties, and surface state density, leading to variations of the photophysics in perovskites [J. Phys. Chem. Lett. 2015, 6, 2200; Sci. Adv. 2017, 2, e1600534; J. Phys. Chem. Lett. 2018, 9, 3891,]. Furthermore, it has been reported that the atmosphere influences perovskite performance as non-coordinated atoms in the film surfaces are partially passivated with H₂O and O₂ molecules [Appl. Phys. Lett. 2014, 105, 183902; ACS Nano 2015, 9, 2311]. In the future, it would be interesting to study different atmosphere effects on the domains we observe in this study.

Now we added below sentences:

“All experiments are carried out under ambient conditions and low humidity conditions below 20%. Our results are consistent over several repetitive scans so we expect that there is no degradation during the measurement.”

on page 3 of the revised manuscript.

4. Authors performed PL measurements in bulk perovskite to show there is no phase segregation. Nevertheless, it is still possible for nanoscale segregation that doesn't manifest itself in the macro.

Response: Yes, we agree.

5. How do the authors avoid the injection of charge carriers with bias?

Response: We don't and we have not said that we do. Bias applied to conductive materials directly leads to current flow.

6. Could authors comment on the fact that similar (in magnitude and dynamics) stresses are produced by applying bias or light? It would be needed to provide information regarding the magnitudes of the local electric fields induced in the film by the different excitation sources and why they result in similar stresses (for example see Fig 3A, 2V vs light).

Response: Thanks for pointing this out. We calculated the electric field to check local electric field when applying bias by AFM tip and shining light illumination. When applying a positive bias voltage of +2 V and under light illumination (500 nm film thickness, perovskite layer/TiO₂ layers), the obtained values are 0.4 kV/cm for biasing tip, and 0.22 kV/cm for light illumination respectively. As shown in Figure 3A in the main manuscript, despite lower electric field for the biasing tip, PFM amplitude under light illumination is higher. We suspect that light induced potential is occurring within the whole film [ACS Energy Lett., 2, 950, (2017)] so it affects the entire film thickness whereas the tip bias is only affecting the surface. Nevertheless, the effects are quite comparable considering the numbers above.

7. The application of negative bias gives rise to structural damage induced by migration of organic cations. Is the process reversible and/or accumulative? Several cycles in which positive and negative bias are sequentially applied would provide insight here.

Response: The structural damage at high voltage above -4V is not actually reversible and is also accumulative. This is shown in our previous publication [Adv. Energy Mat. 6, 1600330, (2016)]. The positive bias does not incur any structural damage whereas the damage starts to occur from the negative bias of -4V (accumulative). When positive bias is applied again, the damage still exists. We now mention this in our manuscript on page 9:

“Structural damage induced by large (>-4V) negative bias is irreversible in our previous report. [Adv. Energy Mat. 6, 1600330, (2016)]”

To check whether there is a reversible effect in the lower voltage range we used, cyclic testing (please see figure R6) is performed consecutively from 0V to +1 V, +2 V, 0 V, -0.5 V, -1 V, and then 0 V for five times regarding amplitude signals as shown in Figure R6. The effect is not accumulative and gives rise to reversible results in the range from -1 V to + 2 V.

Figure R6. The series of amplitude images with applied positive and negative bias for cyclic testing in PFM measurement (a), and average amplitude plot during each cycle (b)

8. The conclusions have important implications when correlating them with solar cells. In this regard, the authors should comment on any evidence for the existence of the stripes in the complete stack –it is well known that after the deposition of the different layers, solvent and interface effects can modify the surface of the perovskite film-.

Response: We agree that there should be some effect of solvents on the surface of the perovskite film which can modify the surface properties. Due to the limitation of the AFM technique which can only probe the top surface, it is almost impossible to determine the structural variation underneath a certain layer deposited on top. However, we do still claim that the stress exists within the film under the light and bias based on our XRD results and this will change the structure of the film in any form, thus, can influence the device efficiency.

We have now added this aspect to our manuscript as the following (page 15):

“There is a chance that surface modification occurs after deposition of other layers in solar cell devices, so the domains may not be the same as we have observed here. However, based on our XRD results, the strain under light and bias will occur within the film bulk.”

Also, if the application of negative bias results in irreversible damage in the perovskite film, the effect should –see above- be accumulative, can the authors provide subsequent JV curve measurements and comment on that? In addition, does the proposed mechanism explain the fast or/and slow term decay

in efficiency when tracking the maximum power point
(DOI:<https://doi.org/10.1016/j.joule.2018.05.005>)?

Response: For the low bias range below -1 V to + 2 V, as explained above, the effects are reversible. However, for larger bias slow term decay can be visible to the limit of surface damage at highest biases. From subsequent J-V results at intermediate bias (Table R1), it can be seen that the efficiency keeps dropping gradually, suggesting slow decay. In fact, the suggested reference and many papers showed that perovskite solar cells often degrade under continuous light soaking and biasing and also show “partial recovery” behaviour after storing the cells in dark but the PCE still does not return to its original value. This indicates that there is irreversible degradation and this could be due to the accumulative effects of ion migration that damage the surface observed in our results.

Time	Jsc (mA/cm ²)	Voc (mV)	FF(%)	Efficiency (%)
1st measurement	23.36	1022.21	76.84	18.35
2nd measurement	23.31	1021.35	76.64	18.25
3rd measurement	23.36	1021.83	76.27	18.21
4th measurement	23.34	1021.19	76.02	18.12

Table R1. J-V results for several consecutive measurements at intermediate bias.

Other comments are:

- In line 124, authors claim that the increase in amplitude is abrupt instead of a gradual increase over time. As far as I understand, there is no timescale in that panel.

Response: This means that the change of signals occurs immediately as soon as light is turned on (within 1s, the time for one scan line in the image).

- What is the light intensity locally? Arbitrary units are not very informative.

Response: Light intensities for all measurements can be found in the experimental section and are 0.3-0.5 W/cm².

- CPD is not defined.

Response: CPD stands for contact potential difference. This definition has been added.

- Lines 203 and 205, I think they refer to Figures S5 and S6 instead.

Response: We have changed Figure S5 → Figure S6.

Reviewer #3 (Remarks to the Author):

The major claim of the paper is that ferroelastic domains form in perovskites upon illumination and/or under applied electrical bias. The authors observe from several microscopy techniques the formation of corrugated features in the perovskite materials under illumination and/or electrical bias. The authors then use X-ray diffraction to observe changes in the structure of the materials as a function of electrical bias and illumination. The results are interesting and (as far as this reviewer is aware) quite novel. If the analysis of the results and data are correct, the work could be a rather influential paper. However, there are substantial questions that remain unanswered from this work. As well, it is not clear that the author's interpretation of the data is either correct or complete.

Response: We thank the reviewer for the valuable comments on our work. All comments are addressed below.

While it is clear from the microscopy that morphological changes occur when illumination and/or bias are present. It is not clear that this is due to reversible processes. While the lack of reversibility is true for both the microscopy and the XRD, I will focus primarily on the XRD since it more specifically reports on both the chemical composition and structure.

No evidence in the article is reported on the reversibility of the X-ray diffraction vs voltage/illumination. The data suggests that these perturbations (light and bias) are causing irreversible chemistry and structural rearrangements to occur. When a bias is applied, a visibly broadened peak appears at 38 degrees and a new peak appears at 45 degrees. Both peaks are rather broad suggesting limited crystallinity of the resulting material. As well, a baseline shift and overall decrease in signal intensity is also observed with applied bias; this is rather characteristic of the formation of amorphous material. This indicates that the perturbations are damaging the samples. It is presently unclear and the authors must convincingly demonstrate, that their data relates to their samples and not the damaged material. Some of the XRD data is difficult to completely assess in Figure S8. I suggest that the authors re-plot Figure S8 so that the data is clearer. Perhaps they can overlay 0V on different potentials in diffractograms or alternatively zoom in on the peaks that they are analyzing.

Response: We have performed additional XRD experiments to check reversibility. Looking at the results in detail, we think that light and bias can cause both reversible and irreversible chemistry and structural rearrangements, depending on the magnitude of applied bias or light intensity. [Adv. Energy Mater., 6, 1501803, (2016)] We now give additional details on specific ranges below and discuss their specific effects on the material. Based on the reviewer's suggestion, we have also re-plotted the XRD patterns for clear illustration (please see Figure R7). Also, we now have added new XRD data after all the measurements in order to compare the reversibility.

Firstly, we take a closer look at our XRD data with the broadened peak near 38 degrees (please see Figure R8). Previously, Hoke et al showed that such peak broadening under the light originates from phase segregation and it is mostly reversible after dark condition for 5 mins. [Chem. Sci., 6, 613, (2015)]. For our sample, it was almost reversible after the illumination, and partially irreversible for applied bias. This could be due to permanent damage caused by a strong ion migration.

Next, we move on to the new peak near 45 degrees (please see Figure R9). It can be seen that the peak does not change under light, however, the new peak extends from 43 degrees (300) and has a maximum at around 44 degrees. Similar results are observed from reversible phase segregation reports [Chem. Sci., 6, 613, (2015)] which stated that the strain is driven by phase segregation under light illumination, which can cause peak splitting and broadening. [Chem. Sci., 6, 613, (2015); ACS Energy Lett., 1, 1199, (2016)] Consequently, we do not think that changed peaks are caused by permanent damage of the material.

We have also done additional XRD measurements to see if there is any damage due to bias and light. If there was an irreversible damage or amorphisation, there should be an increase for the PbI_2 peak near around 13 degrees [Energy Environ. Sci., 7, 2934, (2014)], because halide perovskites usually decompose into MAI or FAI, and PbI_2 [Adv. Funct. Mater. 28, 1705363, (2018)]. A magnified view of the PbI_2 peak shown in Figure R10, the PbI_2 peak does not show noticeable change, however, the intensities of the presented perovskite peaks significantly change, which indicates that the crystallinity of the film has been partly modified due to the perturbations. This could potentially be originating from mostly reversible phase segregation (after dark condition for 5 mins [Chem. Sci., 2015, 6, 613–617]) or reversible conversion mentioned in earlier reports [Adv. Energy Mater. 2016, 6, 1501803]. Therefore, we think that it is not possible to conclude a complete transition to the amorphous phase or irreversible damage to the sample.

Based on above results, we conclude that the film is not permanently damaged by the perturbations, however, there is partial irreversible/reversible chemistry or structural rearrangement, which could originate from phase segregation or externally stimulated ion migration through the material by bias or light. We would like to note that we see damage to the material only at a certain high bias range, especially for above -4 V [Adv. Energy Mat. 6, 1600330, (2016)]. All our measurements are however only performed at lower bias where such damage can be avoided. In fact, this observation can be linked to the structural changes we observed in Fig.2.

Now, we clearly addressed above discussion on page 9 of the revised supplementary information.

Figure R7. Overlay of all the XRD patterns.

Figure R8. Zoom-in of the peaks near 38 degrees.

Figure R9. Zoom-in of the new peak near 45 degrees.

Figure R10. Zoom-on the PbI_2 peak near 13 degrees.

I do not believe that the authors can effectively use Williamson-Hall analysis on their data. The lines that they fit in figure S9 are very loosely correlated with the data that they have. The fit lines are far from unique solutions and cannot with any reliability report on strain in the materials. Part of the problem is related to the number of data points that the authors include, but mostly this is related to the lack of precision in the data. Without clear correlations in the data, the Williamson-Hall analysis and conclusions related to it need to be removed from the manuscript.

Response: In order to provide an improved analysis, we have carefully re-measured and re-analysed XRD peak broadening, and also have repeated the measurements for the Williamson-Hall plot as suggested by the reviewer.

In Williamson-Hall analysis, pure strain broadening is expected to result in a straight line through the origin without intercept [Acta Metall., 1953, 1, 22]. In principle, a smooth line indicates isotropic broadening of each peak regardless of the plane orientation. [Acta Metall., 1953, 1, 22]. However, this is not observed in our results, which means that our film undergoes anisotropic peak broadening. In this case, it is more appropriate to use the broadening functions of separate effects with families of higher order reflections. [J. Appl. Phys. 1961, 32, 2428; Appl. Phys. Lett., 1996, 69, 3173; Phys. Sol. A, 1999, 171, 425]. Previously, it has been reported that halide perovskites have anisotropic strain depending on the crystallographic orientations. [ACS Energy Lett., 2016, 1, 1014; J. Phys. Chem. Lett., 2016, 7, 3683].

With this in mind, we have re-analysed the strain data considering families of strongly oriented (110) and (100) lattice planes using XRD software (X'pert Highscore plus). Based on above arguments and references, we exclude peaks at 24, 38 and 44 degrees because they are not the same family as (100) or (110), and also because they might have contributions from phase segregation as discussed above. As a result, our data has become much more clear and we obtained R-squared (Regression-squared, R^2) values of well above 90% for our linear fits to the data in Williamson-Hall analysis, confirming very reliable data fits. Williamson-Hall analysis for the (110) lattice plane and (100) plane with light/bias perturbations is shown in Fig R11-12. As can be seen in Fig R13, the trend in strain disorder is the same as we described in our manuscript although the absolute values are slightly changed.

Now we have added the above as additional explanation on page 10 of the revised support information.

We believe that this new analysis and data very strongly support our previous claims, and have added these results to the revised manuscript.

Figure R11. Williamson-Hall plot of the XRD peak full width at half maximum (B)

with the light effect

Figure R12. W-H analysis with applying bias

Figure R13. Light and bias-induced strain disorder

Overall, I don't think that the authors can rule out phase segregation or transformation from crystalline to amorphous phases. The changes in morphology could be superficial and merely represent the upper layers of the material, while the majority of the bulk remains unchanged. As well, while the authors do not observe substantial changes at the grain boundaries, it is not necessarily so that phase segregation happens at the grain boundaries. XRD shows the formation of new peaks upon illumination and with applied bias. The XRD also shows a shift in the baseline of the pattern and general decrease in signal intensity of the peaks, which looks like the formation of amorphous material.

Response: This is a very good point. Based on the above results and conclusions made from the new XRD results and analysis, we have added discussion about the possibility of phase segregation and changes in crystallinity of the film due to the perturbations in the revised manuscript. In fact, both effects can be link to the changes in the strain observed in W-H plot and this are also related to the appearance of the ferroelastic domains observed in the PFM measurement.

Now we added below sentences:

“We confirmed that there is no irreversible damage or amorphisation during the measurement as the PbI_2 peak does not change significantly and return to the almost original pattern after applying the stimuli (Fig. R10)”

“In particular, we zoomed the peak at around 37.5 degrees that broadened the most and found that the peak returns to the original state only after the illumination but not after the bias (Fig. R8). Also, the peak at around 44 degrees (Fig. R9) does not change under the light, however, the new peak extends from 43 degrees (300) and has a maximum at around 44 degrees. Such reversible and irreversible variation of the peak is responsible for both reversible and irreversible chemistry and structural rearrangements, depending on the magnitude of applied bias or light intensity which can induce the observed peak broadening.[Adv. Energy Mater., 2016, 6, 1501803; ACS Appl. Mater. Interfaces, 2017, 9, 34970] Also, the peak broadening can be originated from reversible phase segregation [Chem. Sci., 2015, 6, 613] or reversible electric field induced phase conversion. [Adv. Energy Mater., 2016, 6, 1501803; ACS Appl. Mater. Interfaces, 2017, 9, 34970]”

on page 12 of the revised manuscript.

Temperature dependent AFM looks irreversible. I don't think that the authors are looking at the same material after they go through the temperature cycle. One way that the authors could more convincingly show reproducibility would be in Figure 1 b, demonstrating that the amplitude returns to the same level after illumination is turned off.

Response: We have followed the suggestion of the reviewer and now have included data for light off, on and off in PFM scans as below in Fig. S1, which convincingly shows the reproducibility of the effect.

Figure R13. PFM amplitude scans under dark and light illumination (a) and a plot of average amplitude at selected areas (b)

There are several more minor points below:

The authors need to give more details about their light source for illumination experiments. What type of lamp is it? What power density does it produce at the sample location? Was the light filtered? If so, how? What was the approximate spectral distribution? To appropriately assess the amount of energy that the perovskites receive is important for interpreting the possible physical and chemical processes that occur and the conclusions that are drawn.

Response: The used light source and power density for light illumination are a white LED, and 0.3-0.5 W/cm², as measured with an optical power meter. It is not filtered, just white light. For your information, we show spectral intensity depending on wavelength in R14.

We have added this information to the methods section.

Figure R14. Spectral intensity from the used white LED light source.

Perovskites are usually very hygroscopic. What have the authors done to ensure that they do not have water in or on their samples when performing microscopy measurements? The presence of water is known to accelerate degradation pathways under illumination (and possibly under electrical bias as well).

Response: Our all measurements are performed under low humidity condition below 20%. Also, if they are not used for the measurements, the samples are kept in nitrogen. The film surfaces always remain in the same conditions for several weeks while we are performing experiments. Also, we could get consistent results after several measurements unless we intentionally damage the film by applying very high bias.

There are too many data points for the author's photoluminescence measurements vs time. It is difficult to see what is happening to the signal as a function of time. The authors either need to show fewer spectra or plot integrated PL intensity as a function of time.

Response: As shown in Figure R15, we have re-plotted the data with a fewer spectra and changed Fig.S1. Also, we have now added PL peak positions and intensity of all the peaks for clearer illustration (please see Fig. R17)

Figure R15. Time-evolution of photoluminescence under constant illumination

Starting on page 3, last paragraph: The authors should label the section results and discussion. The discussion in the article is disappointingly short, one paragraph on page 15. Considering the amount of experiments that have been performed, surprisingly few points are discussed. It sounds more like the conclusion than a complete discussion, which in part is true considering that significant discussion occurs in the preceding section.

Response: Thanks for the suggestion. We have labelled the results and discussion sections. We have also expanded the discussion section significantly.

Reviewer #4 (Remarks to the Author):

The manuscript provided fundamental understanding of underlying mechanisms related to light and bias induced effects at the nanoscale in hybrid perovskite materials. Particularly, structural variations of the perovskites induced by light and bias are systematically investigated using scanning probe microscopy techniques. It was reported that periodically striped ferroelastic domains, and strain disorder were induced by these applied external stimuli. The structural emergence of domains can affect the transfer pathways for the carriers in general. The findings point to potential origins of I-V hysteresis in halide perovskite solar cells and provide fresh microscopic insight into external stimuli-induced structural disorder governing the superior performance of the metal halide perovskites solar cells. Overall, it is an interesting study regarding the ferroelastic, strain property and carrier behavior from the microscopic point of view. However, some important issues still need to be addressed before it could be further considered. The comments are listed below.

Response: We thank the reviewer for the positive comments on our work. We have addressed all comments below.

1. At present, the meso-scale expression of the ferroelastic and ferroelectric properties in hybrid perovskite materials is the formation of domains. But ferroelasticity is a structural transformation caused by force and ferroelectricity is a synergistic structural transformation caused by electrical polarization. In terms of physical nature, it's different and extremely easy to confuse between the two concepts. The author defines the light-induced domains as ferroelastic domains while these domains still be regarded as ferroelastic domains in the bias-induced part. This requires further discussion related to the physical origin of domain formation under different stimulus conditions to distinguish them. In addition, we see that the authors established the AFM characterization quite well, and therefore is it possible to dig out the mathematical relationship (linear and non-linear) of the AFM signal from the physical point of view?

Response: We agree with the reviewer regarding the description of 'ferroelasticity' and 'ferroelectricity'. The confusion noted by the reviewer was quite apparent in earlier published work, e.g. these well-aligned stripe domains were referred to as ferroelectric in some [Energy Environ. Sci., 10, 950, (2017); PNAS 114, E5504, (2017); J. Phys. Chem. Lett., 6, 1155, (2015)], while ferroelastic in other publications [J. Phys. Chem. C, 120, 5724 (2016); Appl. Phys. Lett., 113, 072102, (2018)]. However, this question has now been settled unambiguously through a range of complementary experimental data/techniques showing that these well-aligned stripe domains are indeed ferroelastic [Sci. Adv., 3, e1602165 (2017); Nat. Commun., 8, 14547, (2017); J. Phys. Chem. C, 120, 5724, (2016)]. These ferroelastic stripe domains arise because of the coexistence of different orientations of the centrosymmetric tetragonal crystal structure [Phys. Rev. B., 2, 754, (1970); Nat. Commun., 8, 14547, (2017)]. Therefore, these well-aligned stripe domains in our investigation are ferroelastic, and

consequently do not show electrically-switchable hysteretic piezo-electric behaviour consistent with earlier studies [Nat. Mater., 14, 193, (2015); J. Phys. Chem. C, 120, 5724, (2016)]. Nonetheless, this ferroelastic domain structure couples indirectly to an applied external stimulus, which can be in the form of an optical illumination or the electrical bias. The applied external stimulus causes ion migration through the sample and can lead to generation of stress or stress gradients of sufficiently high magnitudes. These elastic gradients then in turn can influence ferroelastic domains.

With regard to “mathematical relationship (linear and non-linear) of the AFM signal from the physical point of view”, it has already been established for this well-ordered striped ferroelastic domain structure in Ref. [J. Phys. Chem. C, 120, 5724, (2016), Figure S1]. The relationship is linear for applied biases below approx. 2.5 V and thereafter it becomes non-linear.

To address the point raised by reviewer, we explicitly note the following

“These well-aligned stripe domains in our investigation are ferroelastic, and consequently do not show electrically-switchable hysteretic piezo-electric behaviour consistent with earlier studies [J. Phys. Chem. C, 120, 5724, (2016); Nat. Commun., 8, 14547, (2017); Sci. Adv., 3, e1602165, (2017)]. However, the ferroelastic domain structure couples indirectly to an applied external stimulus, which can be in form of an optical illumination or the electrical bias. The applied external stimulus causes ion migration through the sample and can lead to generation of stress or stress gradients of sufficiently high magnitudes. These elastic gradients then in turn can influence ferroelastic domains.”

on page 9-10 of the revised manuscript.

2. The author attributed the light-induced ferroelastic domains to light-driven molecular reordering in line 112 and further revealed that the origin of the domain patterns is related to a cubic to tetragonal phase transition during film processing in line 159. We agree that the existence of the ferroelastic properties of MAPbI₃ is due to the asymmetrical tetragonal phase structure induced MA molecule, however currently, it is generally accepted that the perovskite of the (FAPbI₃)_{0.85}(MAPbBr₃)_{0.15} composition system is the cubic phase structure. Therefore, we would like to know whether there are also light- and bias-induced domains in pure FAPbI₃ or FA dominating materials systems in order to verify the role of MA molecules.

Response: As far as we are aware, light- and bias-induced stripe domains in pure FAPbI₃ or FA materials systems have not been reported.

3. Further, the author claimed that the appearance of the domain patterns is dependent on the specific grain due to the non-uniform elemental distribution in line 174. Whether the grains with domain are

all tetragonal structures? The composition is rich in MA or even pure MA? Probably TEM data and micro-zone PL could help to prove this point.

Response: As suggested by the reviewer, we performed confocal photoluminescence microscopy with different detection wavelength (650nm and 750nm) in order to check for potential compositional variations between the grains. As can be seen in Figure R15, grains show detection wavelength dependency which implies that there are variations in band gap that could be due to variation of defect type or density.

Previous studies on local phase segregation of $(\text{FAPbI}_3)_{0.85}(\text{MAPbBr}_3)_{0.15}$ showed that there are compositional inhomogeneities across the film and there is structural and compositional inhomogeneity across the film at the nano-meter scale which cannot be detected by standard tools such as XRD or Raman spectroscopy [J. Am. Chem. Soc., 138, 15821, (2016)]. The study has shown that there is a variety of differences in composition including a pure MAPbI_3 phase and mixed phases such as $\text{FA}_y\text{MA}_x\text{PbI}_x\text{Br}_{3-x}$ and $\text{FA}_x\text{MA}_y\text{PbI}_3$ perovskite phases. Also, compositional variation exists in single cation perovskite (MAPbI_3) films as well. We therefore conclude that each grain would not have exact identical composition, which indicates that the strain in each grain could be slightly different.

Figure R15. Confocal microscopy images measured with excitation wavelength of 550 nm at the detection wavelength of 750 nm (a) and 650 nm (b)

4. It is known that MAPbI_3 has phase transition from tetragonal to cubic phase near 57 °C. However, the similar transition occurs in the mixed cation perovskite at about 72 °C is not reported. Therefore, it requires the DSC (differential scanning calorimetry) data of the $(\text{FAPbI}_3)_{0.85}(\text{MAPbBr}_3)_{0.15}$ composition. If such phase transitions occur only in certain grains rather than in the entire polycrystalline film, the question is whether phase transitions in different crystal grains have different transition temperatures due to compositional non-uniformity. I think that the author should make the point clear in the text whether this phase change is partial or integral.

Response: DSC data of $(\text{FAPbI}_3)_{0.85}(\text{MAPbBr}_3)_{0.15}$ has been reported previously [Nature, 517, 476, (2015)] and no information can be found about the phase transition. As stated above, the $(\text{FAPbI}_3)_{0.85}(\text{MAPbBr}_3)_{0.15}$ has composition difference at the nanoscale, thus the phase transitions in different crystal grains have different transition temperatures due to compositional non-uniformity, i.e. phase change is partial within some grains.

We now have added below sentence on page 6:

“We expect that phase transitions in different crystal grains have different transition temperatures due to compositional non-uniformity. Therefore, not every grain should have the phase transition or the same phase transition temperature.”

5. Although various phenomena (phase segregation, photo-induced lattice expansion, and ion migration) under illumination will affect the experimental results to some extent, we believe that it is reasonable to associate the light-induced domain with the ferroelastic phenomenon. However, the photoluminescence (PL) spectral measurements over several minutes in Fig S1B cannot support the absence of ion migration. The authors may need to fit the data to provide the peak position and FWHM changes over time.

Response: We plotted peak position and FWHM and clearly see a red shift against time. Although this does not support the phase segregation observed by Hoke et al, there is a possibility that the phase is being changed slowly over time.

We have now put this in the revised support information.

Figure R17. Variation of peak positions and FWHM in photoluminescence spectral measurement as a function of time

6. We know that AFM (including KPFM, PFM) is a very powerful surface characterization technique. Meanwhile, we appreciated the author by providing many exciting data to prove the domain structure changes. However, regarding the entire manuscript, most analysis the author provided are related to the film surface. Only XRD data has been presented for the structural characterization of the film bulk. However, I think the XRD data is not sufficient to support the structural variations that caused by the light- or bias-induced domain. This is because that the influence of ion migration cannot be ignored in this work, particularly the migration of halide ions I and Br under light and bias, despite that the larger Pb ions are difficult to migrate. Thus, it is recommended that authors could provide more data to exclude the interference of the ions migration.

Response: This is a good point. Regarding studies of the film surface, we would like to point out that the stripe domain structures manifest as surface corrugations, however, they can not exist without also being present in the bulk [Nat. Commun. 8, 14547, (2017)]. We note that we do not rule out ion migration. In halide perovskite, under external stimuli such as light and bias, ion migration always exist which is inevitable at room temperature.

We have added below discussion on page 5:

“To further examine the effects on ion migration on the PFM signal, we examined the PFM signal before and after the light illumination. Fig. R13 shows that PFM signals return to the initial dark values as soon as light illumination is turned off. These results imply that the ion migration does not have direct effect on the above observed PFM results. However, we cannot completely rule out the effects of ion migration which also occurs within the film and change the octahedral distortion which affects the stress within the film. [Phys. Rev. Materials, 2018, 2, 025401]”

7. There are two errors about the perovskite composition in Figure 1A and the definition associated with J-V scan direction in line 302.

Response: We have fixed these errors:

$(\text{FAPbI}_3)_{0.75}(\text{MAPbBr}_3)_{0.15} \rightarrow (\text{FAPbI}_3)_{0.85}(\text{MAPbBr}_3)_{0.15}$ in Figure 1A,

and I-V \rightarrow J-V

8. The author proposed a mechanism that such structural variation under light and bias can cause I-V hysteresis. However, it was documented that many high efficiency devices exhibit almost no

hysteresis by tuning perovskite materials, device architecture, or the transport materials. Therefore, the significance of the current results still needs to be further explored.

Response: The I-V hysteresis is a very common phenomena in halide perovskite solar cells and it is true that the hysteresis is less when the device efficiency is high. However, this hysteresis is highly dependent on the sweeping rate and there is no such device that has almost no hysteresis at high sweeping rate such as 0.1V/s that we have used in our study. If we use low scan rate, we also see almost no hysteresis. In order to mitigate ion conduction and hysteresis, one needs to understand the underlying reasons, which is what we have explored in this work.

Reviewers' comments:

Reviewer #1 (Remarks to the Author):

The authors have provided substantially new evidence that fully addressed my comments and concerns. I can now recommend its publication in Nature Communications.

Reviewer #2 (Remarks to the Author):

The authors have convincingly addressed the major points raised in the review process and the manuscript is definitely improved. Therefore, I recommend the publication of the manuscript in Nature Communications without further revision.

Reviewer #3 (Remarks to the Author):

The authors have addressed many of my original concerns. The new Williamson-Hall analysis is much more convincing. The PL data appears to suggest phase segregation; broadening of the emission peak is a good example of emission coming from multiple sources. This is in line with the authors XRD data. I agree with the authors that there is a precedent for some phase segregation in perovskites to be reversible; however, that was not observed in the data relating to the electrical bias for this article. The XRD does not show that the material is the same after having an electrical bias applied. Peak intensity decreases for all of the peaks associated with the perovskite and they do not recover when returning to the original potential. If the material becomes amorphous or less crystalline during electrical bias, one would not expect to see the formation of peaks from lead(II) iodide as the authors suggest. Amorphous materials do not diffract X-rays and their formation would simply lead to a loss of signal intensity, precisely as observed.

In the modified article the authors state...

"We confirmed that there is no irreversible damage or amorphisation during the measurement as the PbI₂ peak does not change significantly and return to the almost original pattern after applying the stimuli (Fig. R10)"

This does not appear to be correct based on the XRD data. The authors need to address that some irreversible chemistry has occurred due to applied bias or show that they have same material after applied bias. The overall conclusions of the article are probably sound; domains form upon light and electrical bias and they return to some extent to their original positions after the biases are removed, but it would be irresponsible not to discuss that some of the material probably is not the same as before the measurement. An example of how the authors could convincingly show that they have the same material would be to cycle between biases several times and see no change in the structure.

Reviewer #4 (Remarks to the Author):

I believe the work has been improved. I am glad to see the new data has been provided to confirm these well-aligned stripe domains ferroelastic, and consequently do not show electrically-switchable hysteretic piezo-electric behavior. And the authors performed additional XRD experiments to check the structural variations reversibility. However, there is a key point that need to be figured out before published in Nature Communications.

The hysteresis effect in perovskite solar cells is affected by many factors, such as the device

configuration, the defects originated from perovskites or interfaces, the carrier mobility of the transport layer cannot match, etc. How to eliminate these factors and independently relate structural variations to hysteresis behavior is the point that the authors should focus on.

To show the reference device performance, the authors should provide statistics parameters on the hysteresis behavior at different sweeping rate. If the reference device is not good enough, I think it is meaningless to use these devices to explain the structure variables and hysteresis behavior, because the author has no way to rule out the interference of other factors.

At present, I believe that the relationship between strain disorder caused by light and bias stimulation and the hysteresis effect is not fully clear, and there is not enough evidence to prove that they have inevitable dependence. A lot of work has been shown that the high efficiency devices have almost no hysteresis under the light in some reports or certification for high efficiency device. Therefore, it is necessary to rethink the role of structural variables in device performance to provide more data to prove the author's point of view.

In addition, the UPS (ultraviolet photoemission spectroscopy) measurement under light and bias may help the author explain the electronic structure change induced in the stripe domains.

Reviewer #1's comments (Remarks to the Author):

The authors have provided substantially new evidence that fully addressed my comments and concerns. I can now recommend its publication in Nature Communications.

Response: We thank the reviewer for the comment.

Reviewer #2's comments (Remarks to the Author):

The authors have convincingly addressed the major points raised in the review process and the manuscript is definitely improved. Therefore, I recommend the publication of the manuscript in Nature Communications without further revision.

Response: We thank the reviewer for the comment.

Reviewer #3's comments (Remarks to the Author):

The authors have addressed many of my original concerns. The new Williamson-Hall analysis is much more convincing. The PL data appears to suggest phase segregation; broadening of the emission peak is a good example of emission coming from multiple sources. This is in line with the authors XRD data. I agree with the authors that there is a precedent for some phase segregation in perovskites to be reversible; however, that was not observed in the data relating to the electrical bias for this article. The XRD does not show that the material is the same after having an electrical bias applied. Peak intensity decreases for all of the peaks associated with the perovskite and they do not recover when returning to the original potential. If the material becomes amorphous or less crystalline during electrical bias, one would not expect to see the formation of peaks from lead (II) iodide as the authors suggest. Amorphous materials do not diffract X-rays and their formation would simply lead to a loss of signal intensity, precisely as observed.

In the modified article the authors state...

“We confirmed that there is no irreversible damage or amorphisation during the measurement as the PbI₂ peak does not change significantly and return to the almost original pattern after applying the stimuli (Fig. R10)”

This does not appear to be correct based on the XRD data. The authors need to address that some irreversible chemistry has occurred due to applied bias or show that they have same material after applied bias.

Response: Thanks for pointing this out. We agree that amorphous material would lead to a loss of signal intensity. In order to clarify this point, we compared the peak intensity of major peaks, (100) and (110) and added additional discussion.

Figure R1. Magnified intensities and summarized peak heights of (100) and (110) peaks.

As the reviewer commented, the XRD intensity is decreased for higher applied positive biases compared to the original values and we agree that this is a sign of some amorphisation being present. In fact, such irreversible changes align with irreversible surface damage shown in Fig. 4(c), where high bias voltages are applied to the tip. The above figure has been added to the supplementary information.

We also added the following discussion regarding amorphisation on page 12 as below:

“We compared peak intensities before and after the external stimuli for the first two peaks (13.9 and 19.5 degrees, please see Fig.R1) in order to confirm crystallinity of the sample before and after the external stimuli. Under light illumination, the peak intensities return almost to the original values. However, significant changes are observed after applying bias voltages. For positive voltages, both peaks increase and slightly decrease when 0V is applied again. Then, the peak intensities almost do not vary upon negative voltages and remains the same when 0V is applied again. The reduced intensities after applying voltages suggest that the sample undergoes irreversible amorphisation to some extent (reduced crystallinity) after the bias voltages.”

Also, we modified below sentence on page 12 that the reviewer pointed out:

“We confirmed that there is no major damage during the measurement as the PbI_2 peak does not change significantly (PbI_2 being one of the expected degradation products reported before (Adv. Energy Mater. **6**, 1501803, 2016) and return to the almost original pattern after applying the stimuli (Fig. R10). We note however, that some extent of amorphisation of the material could be present as shown by decreased peak intensities in Fig. S8.”

The overall conclusions of the article are probably sound; domains form upon light and electrical bias and they return to some extent to their original positions after the biases are removed, but it would be irresponsible not to discuss that some of the material probably is not the same as before the measurement. An example of how the authors could convincingly show that they have the same material would be to cycle between biases several times and see no change in the structure.

Response: We hope that above discussion and correction answered the reviewer’s comment.

Reviewer #4's comments (Remarks to the Author):

I believe the work has been improved. I am glad to see the new data has been provided to confirm these well-aligned stripe domains ferroelastic, and consequently do not show electrically-switchable hysteretic piezo-electric behavior. And the authors performed additional XRD experiments to check the structural variations reversibility. However, there is a key point that need to be figured out before published in Nature Communications.

The hysteresis effect in perovskite solar cells is affected by many factors, such as the device configuration, the defects originated from perovskites or interfaces, the carrier mobility of the transport layer cannot match, etc. How to eliminate these factors and independently relate structural variations to hysteresis behavior is the point that the authors should focus on.

To show the reference device performance, the authors should provide statistics parameters on the hysteresis behavior at different sweeping rate. If the reference device is not good enough, I think it is meaningless to use these devices to explain the structure variables and hysteresis behavior, because the author has no way to rule out the interference of other factors.

Response: In Henry Sanith's recent paper on hysteresis (ACS Energy Lett. 3, 2472–2476, 2018), it is stated that

“By now, it seems most conceivable that hysteresis is an epiphenomenon, caused by the presence of both mobile ionic species and trap assisted and/or surface charge recombination.”

“Metal halide perovskites are partially ionic in nature. It therefore seems likely that, even despite breakthrough discoveries, ionic movement throughout the absorber material will be with us for the foreseeable future.”

This clarifies that there can be reduced or less hysteresis but there cannot be “no hysteresis” unless the material is made free of defects.

There are not many labs which hold a certified efficiency of perovskite solar cells. We obtained a certification of 1cm² perovskite solar cells (ACS Energy Letters 2, 1978-1984, 2017) from New Port Certification Centre, which confirms that our devices are good enough for studying the fundamental aspects of this material. If we use a slower scan rate of 0.1V/s, the I-V hysteresis of our devices is negligible (ACS Energy Lett. 2, 438–444, 2017)

At present, I believe that the relationship between strain disorder caused by light and bias stimulation and the hysteresis effect is not fully clear, and there is not enough evidence to prove that they have inevitable dependence. A lot of work has been shown that the high efficiency devices have almost no hysteresis under the light in some reports or certification for high efficiency device. Therefore, it is

necessary to rethink the role of structural variables in device performance to provide more data to prove the author's point of view.

In addition, the UPS (ultraviolet photoemission spectroscopy) measurement under light and bias may help the author explain the electronic structure change induced in the stripe domains.

Response: Please note that our paper is not focusing on revealing “relationship between strain disorder caused by light and bias stimulation and the hysteresis effect is not fully clear”. Rather, we aim to explore the effects of structural variation due to the external stimuli.

Also, as the reviewer mentioned above, there are various reasons for the I-V hysteresis. So far, there has been no report on a possible cause of the hysteresis due to the structural variations. Therefore, we believe that our suggested model (Fig.5) will provide new insight into the hysteresis behaviour, which we expect to further detailed findings.

REVIEWERS' COMMENTS:

Reviewer #3 (Remarks to the Author):

The authors have addressed all of my concerns regarding the formation of amorphous material. I find the work very satisfying and an excellent contribution to the literature.

Reviewer #4 (Remarks to the Author):

The authors have addressed my concern, and it is recommended for publication.